# Benchmarking End-To-End Performance of AI-Based Chip Placement Algorithms

**Zhihai Wang**[1]* **Zijie Geng**[1]* **Zhaojie Tu**[1]* **Jie Wang**[1]† **Yuxi Qian**[1] **Zhexuan Xu**[1]
**Ziyan Liu**[1] **Siyuan Xu**[2] **Zhentao Tang**[2] **Shixiong Kai**[2] **Mingxuan Yuan**[2]
**Jianye Hao**[2,3] **Bin Li**[1] **Feng Wu**[1]

[1] University of Science and Technology of China     [2] Noah's Ark Lab, Huawei     [3] Tianjin University
{zhwangx, zijiegeng, tuzj}@mail.ustc.edu.cn    jiewangx@ustc.edu.cn

## Abstract

Chip placement is a critical step in the Electronic Design Automation (EDA) workflow, which aims to arrange chip modules on the canvas to optimize the performance, power, and area (PPA) metrics of final designs. Recent advances show great potential of AI-based algorithms in chip placement. However, due to the lengthy EDA workflow, evaluations of these algorithms often focus on *intermediate surrogate metrics*, which are computationally efficient but often misalign with the final *end-to-end performance* (i.e., the final design PPA). To address this challenge, we propose to build **ChiPBench**, a comprehensive benchmark specifically designed to evaluate the effectiveness of AI-based algorithms in final design PPA metrics. Specifically, we generate a diverse evaluation dataset from 20 circuits across various domains, such as CPUs, GPUs, and NPUs. We then evaluate six state-of-the-art AI-based chip placement algorithms on the dataset and conduct a thorough analysis of their placement behavior. Extensive experiments show that AI-based chip placement algorithms produce unsatisfactory final PPA results, highlighting the significant influence of often-overlooked factors like regularity and dataflow. We believe ChiPBench will effectively bridge the gap between academia and industry.

## 1 Introduction

The exponential growth in the scale of integrated circuits (ICs), in accordance with Moore's law, has posed significant and increasingly complex challenges to chip design [1, 2]. To address the growing complexity and enhance efficiency, numerous electronic design automation (EDA) tools have been developed to assist hardware engineers . As shown in Figure 1, EDA tools automate various critical steps in the chip design workflow, including high-level synthesis, logic synthesis, physical design, testing, and verification [1, 3].

Chip placement is a critical step in the chip design workflow, focused on determining the locations of chip components within the die to optimize the performance, power, and area (PPA) metrics of the final chip designs [4–6]. Traditionally, this process has relied on manual placement by expert designers, requiring significant labor and extensive domain expertise. To improve efficiency, numerous automation methods, particularly AI-based algorithms, have been developed to streamline this task. These methods can be broadly classified into three categories: black-box optimization (BBO) method, analytical methods (gradient-based methods), and reinforcement learning (RL) methods. Black-Box-Optimization (BBO) Methods, such as simulated annealing (SA) [7] and evolutionary algorithms (EA) [5], treat macro placement as a black-box optimization problem, searching the design space

---

*Equal contribution.
†Corresponding author.

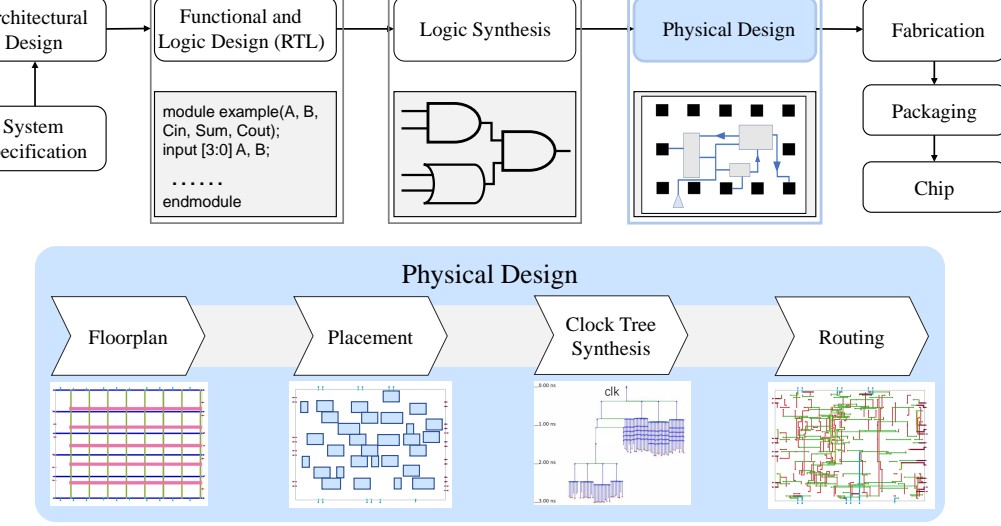

Figure 1: Illustration of the modern chip design workflow.

for near-optimal solutions without explicitly leveraging gradient information. Analytical methods formulate the placement objective as a differentiable function of module coordinates and employ gradient-based or numerical optimization techniques to efficiently solve for macro positions [8, 9]. Recent research has framed macro placement as a Markov Decision Process (MDP), where macro positions are determined sequentially [10, 11]. Machine Learning, such as reinforcement learning (RL) has thus emerged as a promising approach, iteratively improving placement performance by learning from environmental feedback through trial and error [12–16].

Nevertheless, given the extensive process involved in chip design, these algorithms are typically assessed using *intermediate surrogate metrics*. While computationally efficient, such metrics often fail to closely align with *end-to-end performance* (i.e., the final design PPA). On one hand, obtaining the end-to-end performance of a given chip placement solution requires substantial engineering effort due to the lengthy chip design workflow. In particular, we observed that directly applying existing open-source electronic design automation (EDA) tools to certain widely used chip placement datasets often fails to produce reliable end-to-end performance results. On the other hand, since PPA metrics are influenced by numerous factors that are considered in earlier design stages, a critical gap exists between certain intermediate metrics and the final PPA objectives. Consequently, this gap limits the applicability of existing AI-driven placement algorithms in practical industrial scenarios.

To address this challenge, we propose ChiPBench, a comprehensive benchmark designed for EDA tasks, especially for evaluating AI-based chip placement algorithms in terms of their effectiveness in improving final PPA metrics. Appealing features of ChiPBench include its fully open-source and reproducible characteristics, covering the entire EDA workflow from the source Verilog code, and unifying the evaluation framework of AI-based chip placement methods using end-to-end performance. Thus, ChiPBench can effectively facilitate research in chip placement within the AI community by taking the first step toward a fully reproducible, unified evaluation framework. In terms of the dataset, we have generated 20 circuits from various domains (e.g., CPUs, GPUs, and NPUs). Then, these designs are compiled by executing the workflow from the Verilog source code, preserving sufficient physical implementation kits, which enable evaluations of the placement algorithms regarding their impact on the final PPA. For the evaluated algorithms, we executed *six* state-of-the-art AI-based chip placement algorithms on the aforementioned benchmark and *plugged* the results of each single-point algorithm into the physical implementation workflow to obtain the PPA results. Experimental results reveal that even when a single-point algorithm excels in intermediate metrics, its final PPA results can remain unsatisfactory. Visualization experiments further show weak correlations between some intermediate metrics and final PPA, highlighting the need to focus on optimizing final PPA directly. This indicates that we need to explore proxy metrics that more closely reflect physical realism and end-to-end performance as the objectives and features of AI algorithms.

We believe that our benchmark will serve as an effective evaluation framework to bridge the gap between academia and industry.

We summarize our major contributions as follows: (1) Our proposed ChiPBench is a reproducible and unified evaluation framework for existing AI-based chip placement algorithms, utilizing end-to-end performance with fully open-source EDA tools. This can effectively facilitate research in chip placement within the AI community. (2) We construct a new dataset of 20 circuits from various domains, fully generated using an open source EDA pipeline from the source Verilog code. Our specialized procedure addresses missing macros in open-source circuits, enabling the generation of diverse designs while preserving essential physical implementation data for evaluation. (3) We evaluate six state-of-the-art AI-based chip placement algorithms, including the most popular AI-based chip placement methods. (4) Based on our experiments, our analysis reveals that existing AI-based algorithms often produce unsatisfactory final PPA, highlighting the significant influence of often-overlooked factors such as design regularity and dataflow.

## 2   Related Work

**Datasets** Some well-known EDA conferences, such as ISPD and ICCAD, host contests addressing EDA challenges and offer benchmarks with processed data for researchers. However, in the early years (e.g., ISPD2005 [17] and ICCAD2004 [18]), the provided datasets used overly simplified `Bookshelf` formats, which are abstracted versions of the actual design kits. Therefore, we cannot evaluate the final PPA of the placement results on those datasets. Recently, ISPD2015 [19] and ICCAD2015 [20] have offered benchmarks and datasets closer to real-world applications, including necessary netlist, library, and design exchange files, broadening their utility slightly. Nevertheless, they still lack the essential information (e.g., necessary design kits) to run the open-source EDA tools such as OpenROAD [21]. Beyond these conferences, some other datasets have been developed in various directions. For example, the EPFL [22] benchmarks and the larger OpenABC-D [23] dataset concentrated on synthetic netlists, primarily for testing modern logic optimization tools with a focus on logic synthesis. CircuitNet 2.0 [24], on the other hand, shifted the focus towards providing multi-modal data for prediction tasks, enhancing the capability for various prediction tasks through the use of diverse data modalities. Our dataset provides complete files for each case and necessary design kits, such as timing constraints, library files, and LEF files, offering a comprehensive dataset that supports all stages of physical implementation and fosters a more integrated approach to chip design and evaluation.

**Placement Algorithms** Recent advancements in AI technology within the EDA field have led to a variety of AI-based chip placement algorithms. (1) Black-Box Optimization methods. Simulated Annealing [4] provides a probabilistic method for finding a good approximation of the global optimum. Wire-Mask-Guided Black-Box Optimization [5] uses a wire-mask-guided greedy procedure to optimize macro placement efficiently. (2) Analytical methods. DREAMPlace [6] uses deep learning toolkits to achieve over a 30x speedup in placement tasks. AutoDMP [9] leverages DREAMPlace for the concurrent placement of macros and standard cells, enhancing macro placement quality. (3) Reinforcement Learning methods. MaskPlace [14] treats chip placement as a visual representation learning problem, reducing wirelength and ensuring zero overlaps. ChiPFormer [15] employs offline reinforcement learning, fine-tuning on unseen chips for better efficiency. EfficientPlace [11] combines reinforcement learning with tree search to improve sample efficiency for macro placement. MaskRegulate [25] uses RL for placement refinement, enhancing PPA metrics and ensuring design regularity. The evaluation of these algorithms mainly focuses on intermediate metrics. In contrast, we utilized the *end-to-end performance* to evaluate six existing AI-based chip placement algorithms, encompassing a significant portion of mainstream AI-based placement algorithms.

## 3   Background on Electronic Design Automation

Electronic Design Automation (EDA) is a suite of software tools vital for designing and developing electronic systems, primarily integrated circuits (ICs). These tools enable engineers to efficiently transform innovative concepts into functional products, addressing the complexity and demands of modern chip design. EDA optimizes the entire design process from schematic capture to fabrication, reducing time-to-market and enhancing design precision and sophistication. In the chip design workflow, EDA tools support various functions: they perform simulations to verify circuit behavior,

Table 1: Statistics of designs in our benchmark.

| Id | Design | #Cells | #Nets | #Macros | #Pins | #IOs |
|----|--------|--------|-------|---------|-------|------|
| 1 | ariane133 | 167907 | 197606 | 132 | 979135 | 495 |
| 2 | ariane136 | 171347 | 201428 | 136 | 1000876 | 495 |
| 3 | bp_fe | 33188 | 39512 | 11 | 185524 | 2511 |
| 4 | bp_be | 51382 | 62228 | 10 | 293276 | 3029 |
| 5 | bp | 307055 | 348278 | 24 | 1642427 | 1198 |
| 6 | swerv_wrapper | 98039 | 113582 | 28 | 573688 | 1416 |
| 7 | bp_multi | 152287 | 174170 | 26 | 813050 | 1453 |
| 8 | vga_lcd | 127004 | 151946 | 62 | 706931 | 198 |
| 9 | dft68 | 41974 | 56217 | 68 | 226420 | 132 |
| 10 | or1200 | 26667 | 32740 | 36 | 153379 | 383 |
| 11 | mor1kx | 68291 | 81398 | 78 | 394210 | 576 |
| 12 | ethernet | 35172 | 44964 | 64 | 205739 | 211 |
| 13 | VeriGPU | 71082 | 85081 | 12 | 421857 | 134 |
| 14 | isa_npu | 427003 | 548451 | 15 | 2406579 | 93 |
| 15 | ariane81 | 153873 | 180516 | 81 | 894420 | 495 |
| 16 | bp_fe38 | 26859 | 32661 | 38 | 154162 | 2511 |
| 17 | bp_be12 | 38393 | 47030 | 12 | 220938 | 3029 |
| 18 | bp68 | 164039 | 191475 | 68 | 887046 | 1198 |
| 19 | swerv_wrapper43 | 95455 | 110902 | 43 | 560088 | 1416 |
| 20 | bp_multi57 | 127553 | 146710 | 57 | 680748 | 1453 |

execute synthesis to convert high-level descriptions to gate-level implementations, and manage physical layouts to ensure designs can be realized in silicon.

As shown in Figure 1, the EDA design flow includes several key stages [26]: logic synthesis, floorplanning, placement, Clock Tree Synthesis (CTS), and routing. **Logic Synthesis** transforms a high-level circuit description into an optimized gate-level netlist [27–29]. **Floorplan** involves deciding the layout of major components within an integrated circuit, positioning blocks and core components to balance signal integrity, power distribution, and area utilization. **Placement** involves assigning specific locations to various circuit components—including macro blocks and standard cells—within the core area of the chip, following the floorplanning stage. The primary objective of this stage is to strategically place the components to optimize performance metrics such as delay and power consumption while ensuring adherence to design rules [11]. **Clock Tree Synthesis (CTS)** creates a clock distribution network within an IC to minimize those clock effects, and ensure the correct timing synchronization for circuit operation. **Routing** involves creating the physical paths for electrical connectivity between various components on the IC as per the netlist. This stage must handle multiple layers of the chip, manage signal integrity, and meet all electrical and timing constraints [13].

**Chip Placement** The placement process typically consists of three key phases: macro placement, global placement, and legalization (also referred to as detailed placement). (1) Macro placement is a critical very large-scale integration (VLSI) physical design problem that targets the arrangement of larger components, such as SRAMs and clock generators—often called macros. This phase significantly impacts the chip's overall floorplan and essential design parameters like wirelength, power, and area. (2) Following this, the global placement phase addresses the arrangement of the more numerous and smaller standard cells. This phase typically utilizes analytical solvers to secure an optimized configuration that not only minimizes wirelength but also enhances the electrical and timing performance of the chip. (3) Subsequently, legalization phase refines the placement to meet strict design rules. This involves resolving overlaps between cells, aligning them to predefined rows.

## 4  Dataset

In this section, we first discuss the motivation behind our proposed dataset and provide an overall introduction in Section 4.1. Next, we detail the dataset generation pipeline in Section 4.2, and finally,

we present our proposed procedure for creating diverse, macro-rich designs in Section 4.3. Our dataset is publicly available at Hugging Face, enabling open access and facilitating future research.

## 4.1 Motivation

Due to the oversimplification of datasets in early years, there exists a significant gap between these datasets and real-world applications. For instance, the usually used `Bookshelf` format [17, 18] is overly simplified so that placement results given in such format are inapplicable for the subsequent stages to obtain a valid final design. Some later datasets [20] provide the `LEF/DEF` and necessary files for running these stages, but the contained circuits are still limited and they still lack some information for open-source tools like OpenROAD to work. For instance, the library file lacks buffer definitions, which is necessary for the clock tree synthesis phase, and the LEF file has incomplete layer definitions, which hinders the routing phase.

To tackle this problem, we present a comprehensive dataset that captures physical implementation information across the EDA flow. Our dataset includes data from a complete EDA design flow, starting from Verilog and encompassing key stages such as logic synthesis, floorplanning, placement, clock tree synthesis, and routing. It comprises both newly generated designs and processed data from existing datasets, thereby ensuring a broad coverage of realistic scenarios. For each stage, the dataset provides intermediate design data for every case, enabling tasks such as logic optimization, chip placement, and routing in the EDA domain. Furthermore, the dataset spans various domains—such as CPUs, GPUs, and NPUs—and covers a diverse range of sizes, from designs with a few thousand cells to those with nearly a million. Detailed statistics for each case are provided in Table 1.

## 4.2 Dataset Generation Pipeline

Our dataset generation pipeline begins with the collection of Verilog-defined circuit designs as raw data. To process these designs, we use OpenROAD [21], an open-source EDA tool, ensuring full reproducibility of our results and supporting the open-source community. This approach guarantees that all generated data and methodologies are fully open-source and accessible. The pipeline first defines physical implementation parameters, such as timing constraints, cell density, routing layer configurations, and technology choices for the collected Verilog files. Following this, we perform logic synthesis to generate netlists. Using the predefined parameters, we execute subsequent steps in the physical design flow, including floorplanning, placement, CTS, and routing. Intermediate files, such as `LEF/DEF`, are generated at each stage to facilitate various downstream tasks. For example, DEF files obtained during the pre-placement stage are used to evaluate and apply subsequent placement algorithms effectively.

## 4.3 Diverse Design Generation

In a typical synthesis flow, a Verilog file produces a netlist composed exclusively of standard cells. However, if macro blocks (such as memory blocks) are required, they must be explicitly instantiated in the Verilog description, accompanied by the corresponding macro definition files (e.g., LEF and LIB). Since most open-source circuit repositories lack these files, we developed a specialized procedure to address this limitation and generate diverse design outcomes. Our automated flow enables the partition and hardening of specific modules within a Verilog-defined circuit design into macros, producing a new Verilog file alongside the corresponding definition files. These outputs serve as the foundation for the subsequent dataset generation pipeline, enabling the creation of diverse designs with varying macro counts, shapes, and topological netlist structures. All designs are synthesized and implemented using the NanGate45 open-source technology library, ensuring compatibility with standard EDA tools while maintaining openness and reproducibility.

## 5 Evaluation

Although commercial EDA tools like Cadence Innovus and IC Compiler are commonly used for PPA evaluation, they are typically closed-source and expensive, making them less suitable for academic research and reproducibility. In contrast, the open-source tool OpenROAD [21] is becoming increasingly mature and provides a more accessible platform for the research community. However, establishing a open-source, end-to-end evaluation flow remains challenging due to various technical

obstacles (detailed in the Appendix 6.2). In this work, we bridge this gap by providing an open-source evaluation flow based on OpenROAD. Our flow significantly lowers the barrier to the integrated assessment of placement algorithms, including macro placement, global placement, and mixed-size placement, and enables reproducible and extensible PPA evaluation for academic use.

## 5.1 Evaluation Metrics

### 5.1.1 Final Design PPA Metrics

The primary objective of the EDA workflow is to optimize the final PPA metrics, which represent performance, power, and area—three fundamental dimensions used to assess the quality of a chip. Performance is typically evaluated using worst negative slack (WNS), total negative slack (TNS), and the number of violating paths (NVP). Negative slack indicates timing violations, with WNS identifying the most severe violation, TNS quantifying the total accumulated slack violations, and NVP counting the number of paths failing to meet timing constraints. Area refers to the total footprint of standard cells, while power encompasses the total power consumption of the chip, including internal power, switching power, and leakage power. Optimizing these PPA metrics has been a major focus in the industry and is typically approached through expert-designed heuristics.

### 5.1.2 Intermediate Surrogate Metrics

Commonly used intermediate surrogate metrics include congestion, wirelength, half perimeter wire length (HPWL), and MacroHPWL. Congestion measures wire density across chip regions, where excessive congestion can create routing challenges. Although not a direct PPA component, effective congestion management is crucial for manufacturability, making it a relevant evaluation metric in this study. It is typically estimated after CTS but before detailed routing to refine macro placement and routing strategies. Wirelength represents the total length of all interconnections, while HPWL estimates it using the sum of the half-perimeters of bounding boxes enclosing all pins in each net. MacroHPWL further simplifies HPWL by considering only macros. Additionally, Regularity, which reflects the consistency and uniformity of component placement, is characterized using the approach from [25].

## 5.2 End-to-End Evaluation Workflow

We present an end-to-end evaluation workflow for various stages of the EDA flow, as illustrated in Figure 2. To evaluate a stage-specific algorithm, the output from the preceding stage serves as its input, and the algorithm's output is reintegrated into the original design flow. We apply this flow to the macro placement stage in Sec 5.3, and we also evaluate other stages in Appendix E.2. Final PPA metrics provide a comprehensive assessment, avoiding the limitations of isolated stage-specific metrics. This approach facilitates algorithm optimization by ensuring improvements translate into practical chip design enhancements. Our project is open-sourced on GitHub.

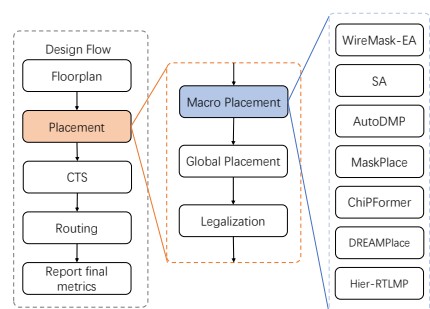

Figure 2: Illustration of our end-to-end evaluation workflow.

## 5.3 Experimental Setup

We apply the proposed workflow to evaluate six macro placement algorithms: SA, WireMask-EA, AutoDMP, MaskPlace, ChiPFormer, and MaskRegulate, along with the traditional algorithm DREAMPlace and Hier-RTLMP [30], which serves as the baseline. A detailed introduction can be found in Appendix A.

As most of these methods only support the circuit data in a `BookShelf` format, while the circuits in our used dataset are in a standard `LEF/DEF`, we start by converting our datasets files to `BookShelf` format to serve as the input for the placement algorithms. After finishing the macro placement stage, the resulting placement files are then converted back to `DEF` format and reintroduced into the original flow. Additionally, we perform global placement and detailed placement using OpenROAD's native

Table 2: The normalized evaluation results of AI-based macro placement algorithms. For each benchmark, the metric of each method is normalized by dividing it by the metric of the baseline (Hier-RTLMP). The final table presents the average of these normalized metrics.

| Method | | Intermediate Metrics | | | | | PPA Metrics | | | | |
|---|---|---|---|---|---|---|---|---|---|---|---|
| | | Placement Metrics | | | Route Metrics | | Timing Performance | | | Power ↓ | Area ↓ |
| | | MacroHPWL ↓ | Regularity ↓ | HPWL ↓ | Congestion ↓ | Wirelength ↓ | WNS ↓ | TNS ↓ | NVP ↓ | | |
| AI-based | WireMask-EA | **0.844** | 1.277 | 1.117 | 1.115 | 1.115 | 1.556 | 11.007 | 3.337 | 1.039 | 1.018 |
| | SA | 0.917 | 1.138 | 1.085 | 1.063 | 1.087 | 1.424 | 2.951 | 1.318 | 1.026 | 1.009 |
| | AutoDMP | 0.925 | 1.257 | **0.892** | **0.941** | **0.950** | 1.282 | 2.599 | 1.208 | 1.021 | 1.008 |
| | MaskPlace | 2.442 | 1.166 | 1.184 | 1.165 | 1.178 | 1.805 | 3.436 | 1.301 | 1.036 | 1.021 |
| | ChiPFormer | 0.901 | 1.357 | 1.097 | 1.123 | 1.119 | 2.616 | 7.624 | 2.162 | 1.037 | 1.013 |
| | MaskRegulate | 1.472 | **0.653** | 1.036 | 1.011 | 1.032 | 1.156 | 2.213 | 1.216 | 1.001 | 1.000 |
| Traditional | DREAMPlace | 1.032 | 1.255 | 1.029 | 1.038 | 1.041 | 2.030 | 7.463 | 1.682 | 1.016 | 1.005 |
| | Hier-RTLMP | 1.000 | 1.000 | 1.000 | 1.000 | 1.000 | **1.000** | **1.000** | **1.000** | **1.000** | **1.000** |

Table 3: The evaluation results of swerv_wrapper under AI-based macro placement algorithms. MacroHPWL($\mu m$), Regularity($\mu m$), and HPWL($\mu m$) serve as metrics during the placement stage. Congestion (%) and Wirelength($\mu m$) are evaluated in the routing stage. WNS (ns), TNS (ns), NVP, Power (nW), and Area($\mu m^2$) are PPA metrics.

| Method | | Intermediate Metrics | | | | | PPA Metrics | | | | |
|---|---|---|---|---|---|---|---|---|---|---|---|
| | | Placement Metrics | | | Route Metrics | | Timing Performance | | | Power ↓ | Area ↓ |
| | | MacroHPWL ↓ | Regularity ↓ | HPWL ↓ | Congestion ↓ | Wirelength ↓ | WNS ↑ | TNS ↑ | NVP ↓ | | |
| AI-based | WireMask-EA | 92304 | 16087 | 5052232 | 0.445 | 6203022 | -1.120 | -1052.140 | 1791 | 0.296 | 235525 |
| | SA | 108068 | 15756 | 4637819 | 0.383 | 5561268 | -1.033 | -863.393 | 1485 | 0.273 | 230076 |
| | AutoDMP | 101651 | 18086 | 4214109 | 0.356 | 5173002 | -0.941 | -903.640 | 1478 | 0.270 | 229290 |
| | MaskPlace | 282636 | 14743 | 4634862 | 0.378 | 5484915 | -0.768 | -582.361 | **1363** | 0.271 | 230706 |
| | ChiPFormer | **89999** | 17512 | 4718772 | 0.408 | 5685641 | -1.352 | -1496.870 | 1537 | 0.277 | 233285 |
| | MaskRegulate | 221155 | **11265** | 3991734 | **0.325** | 4731291 | -0.670 | **-516.377** | 1365 | 0.266 | 228183 |
| Traditional | DREAMPlace | 105719 | 15149 | 3965871 | 0.338 | 4730011 | -0.744 | -572.391 | 1415 | 0.266 | 228845 |
| | Hier-RTLMP | 118198 | 16732 | **3804541** | 0.326 | **4550107** | **-0.660** | -613.774 | 1435 | **0.265** | **226536** |

Place method, completing the entire placement process. Finally, we execute the subsequent flow to obtain end-to-end evaluation results for comparison with other algorithms.

# 6 Results and Discussions

## 6.1 Main Evaluation

We evaluate the AI-based chip placement algorithms, including SA, WireMask-EA, AutoDMP, MaskPlace, ChiPFormer and MaskRegulate, using both intermediate metrics and end-to-end performance. The normalized results for macro placement are in Table 2. The total results from the experiment are detailed in Appendix D; experiments with commercial tools on our dataset and cross-verification between our open-source flow and the commercial toolchain can be found in Appendix E.1. ChiPFormer and WireMask-EA demonstrated a significant reduction in MacroHPWL compared to the baseline algorithm. WireMask-EA achieved the best performance in terms of MacroHPWL. While these AI-based placement algorithms showed good performance on several intermediate metrics, they perform poorly in terms of the PPA metrics compared to traditional algorithm. Although AI-based placement algorithms have achieved significant progress in improving certain intermediate metrics, their impact on enhancing final PPA remains quite limited.

## 6.2 Detailed Correlation Analysis

In this section, we analyze the correlation between intermediate metrics and final PPA metrics in existing placement algorithms. Specifically, we normalize the metrics for each algorithm on each benchmark and compile them into a dataset for correlation analysis. For normalization, we use the results of Hier-RTLMP as a baseline and express all other methods' metrics as relative ratios with respect to it on the same design. We then compute the Pearson correlation coefficient [31] to quantify the strength of the linear relationship between metric pairs. To ensure consistency, we adjust all values so that lower values indicate better

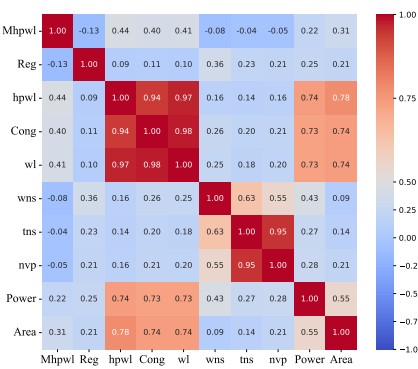

Figure 3: Correlation Between Metrics

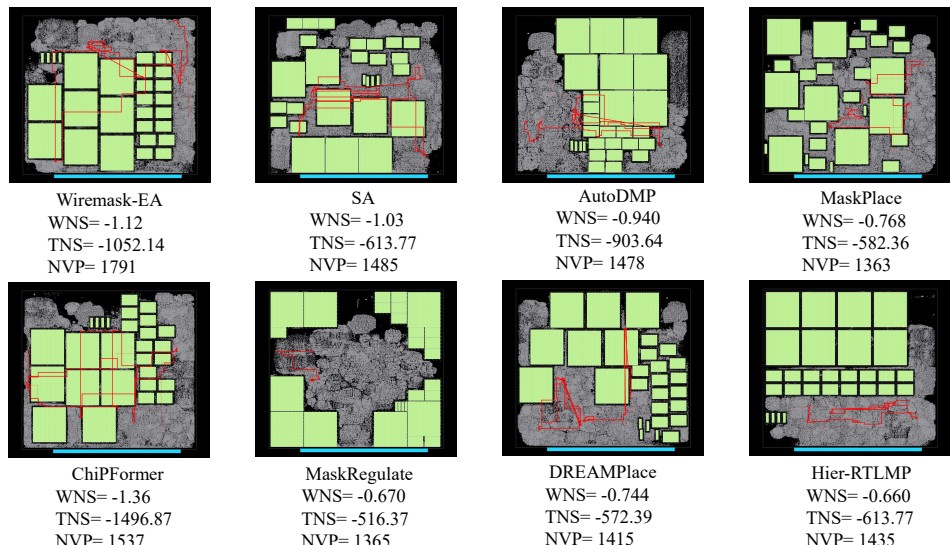

Figure 4: Images of the worst timing path for each method in swerv_wrapper. The results show that ChiPFormer, WireMask-EA, and AutoDMP have the worst timing performance, while MaskRegulate achieves the second-best WNS and the best TNS among all algorithms.

performance across all metrics. The results, presented in Figure 3, reveal key insights into the relationships between intermediate and final PPA metrics.

First, we observe strong correlations among certain intermediate metrics (e.g., HPWL, congestion, and wirelength) and among specific PPA metrics (e.g., TNS and NVP). This suggests that, in certain scenarios, optimizing one key PPA metric (e.g., TNS) could implicitly improve others (e.g., NVP), as TNS approximates the product of average WNS and NVP, with WNS values exhibiting minimal variation across different placement results.

Second, several intermediate metrics also exhibit moderate correlation with final PPA metrics. In particular, HPWL, congestion, and wirelength have a noticeable influence on power and area. This can be attributed to the fact that, during the CTS stage, timing violations are addressed—often by inserting buffers (a type of standard cell) to correct setup and hold violation paths. A larger HPWL can degrade timing slack predictions, necessitating more aggressive buffer insertion and thereby increasing standard cell area and power consumption. For instance, although the WireMask-EA method achieves the best MacroHPWL results, its weak optimization of HPWL, congestion, and wirelength ultimately leads to suboptimal power and area outcomes. The close interdependence of these PPA metrics underscores the necessity of holistic multi-objective optimization rather than focusing on a single metric in isolation.

However, MacroHPWL is an intermediate metric that exhibits a low correlation with final PPA metrics, indicating that it is oversimplified and not well aligned with end-to-end evaluation criteria. Additionally, timing-related metrics (WNS, TNS, and NVP) exhibit weak correlations with other intermediate metrics, suggesting inconsistencies in how existing proxy metrics translate to timing performance optimization. This discrepancy highlights the need for closer attention to the design of proxy metrics to ensure a consistent and effective optimization process for timing performance.

Finally, because obtaining accurate final PPA metrics is time-consuming and computationally expensive, designing effective intermediate metrics is essential. Better alignment of these metrics with final PPA—particularly timing performance metrics such as WNS, TNS, and NVP—could significantly accelerate design space exploration, facilitating more efficient and reliable optimization.

## 6.3 Case Study on Performance Metrics

Building on the preceding correlation analysis, this section presents a case study examining how different placement algorithms and intermediate metrics influence timing performance. We focus on

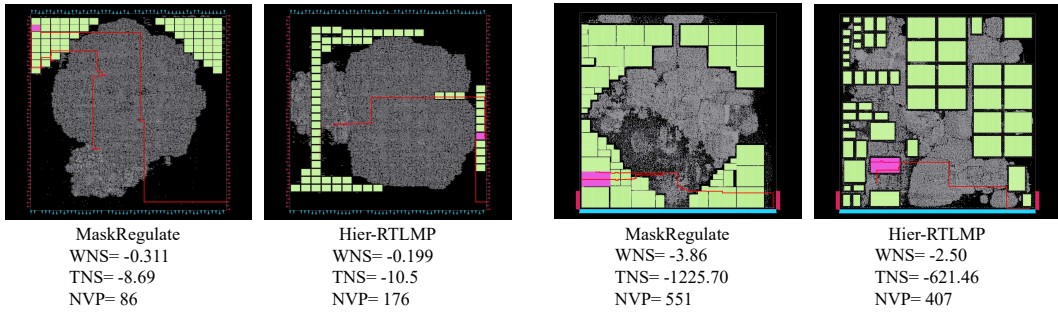

(a) The worst timing path in *ethernet*.

MaskRegulate
WNS= -0.311
TNS= -8.69
NVP= 86

Hier-RTLMP
WNS= -0.199
TNS= -10.5
NVP= 176

(b) The worst timing path in *bp_multi57*.

MaskRegulate
WNS= -3.86
TNS= -1225.70
NVP= 551

Hier-RTLMP
WNS= -2.50
TNS= -621.46
NVP= 407

Figure 5: Comparison of MaskRegulate and Hier-RTLMP: Hier-RTLMP achieves better WNS on *ethernet* benchmark and demonstrates superior timing performance on *bp_multi57* benchmark compared to MaskRegulate.

the *swerv_wrapper* benchmark. Table 3 summarizes the results, and Figure 4 illustrates variations in worst timing paths across different algorithms.

Among all methods, ChiPFormer, WireMask-EA, and AutoDMP achieve the best MacroHPWL scores but exhibit poor timing performance. Their placement results reveal significant macro clustering near the center. While this reduces MacroHPWL by shortening macro-to-macro connections, it pushes standard cells to the periphery and introduces severe routing congestion. Macro clusters consume valuable routing layers, dispersing shared-net standard cells and increasing wirelength. These longer and more complex routing paths result in higher path delays, leading to degraded WNS and TNS. This reflects a clear case of overfitting to MacroHPWL, where optimization of a single intermediate metric fails to translate into improved timing due to physical side effects.

In contrast, the Regularity metric encourages macros to be placed near the boundary, preserving central space for standard-cell clustering and improving routing efficiency. For example, MaskRegulate, which uses Regularity, achieves the best TNS and near-optimal WNS. Its placement enables shorter timing paths and more efficient cell connections. Compared to the traditional Hier-RTLMP algorithm, which also biases peripheral macro placement, RL-based approaches like MaskRegulate demonstrate better adaptability in balancing routability and timing.

Figure 5 offers further insights through two benchmarks. Although MaskRegulate alleviates central congestion via boundary-aware placement, it sometimes overlooks internal dataflow. In certain cases, macro clustering disrupts critical communication paths, resulting in longer wires and degraded slack. In contrast, Hier-RTLMP explicitly considers communication structure and achieves better timing. This underscores the importance of dataflow-aware placement for robust timing optimization in addition to routability improvements.

## 6.4 Discussion

Our analysis exposes a critical gap in AI-based chip placement: while algorithms excel at optimizing the intermediate metrics,they fail to effectively enhance final PPA due to weak proxy correlations and unintended physical side effects. Given the high computational cost of obtaining final PPA, directly incorporating it into AI optimization is impractical. This highlights the urgent need for more advanced insights to bridge the gap between intermediate indicators and final PPA. We identify three promising research directions: (1) designing intermediate metrics that better align with final PPA by organically integrating factors such as regularity and dataflow; (2) developing feature-based surrogate models to approximate final PPA more efficiently; and (3) leveraging multi-fidelity optimization and learning to balance cost and accuracy in AI-based placement algorithms.

## 7 Conclusion

This paper presents a comprehensive dataset spanning the full spectrum of the EDA design process and an end-to-end evaluation method, which we used to assess several placement algorithms: SA,

WireMask-EA, AutoDMP, MaskPlace, ChiPFormer, and MaskRegulate. Our evaluation revealed inconsistencies between metrics emphasized by mainstream placement algorithms and final PPA. These findings highlight the need for a new perspective in placement algorithm development.

## Acknowledgments

The authors would like to thank all the anonymous reviewers for their insightful comments. This research was supported in part by National Key R&D Program of China under contract 2022ZD0119801, National Nature Science Foundations of China grants U23A20388 and 62021001. This research was also supported by the advanced computing resources provided by the Supercomputing Center of the USTC.

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

# A  Algorithms

AI-based chip placement algorithms can be roughly grouped into three categories: black-box optimization (BBO) methods, analytical methods (gradient-based methods), and reinforcement learning (RL) methods. Each category frames the placement task as an optimization problem but adopts distinct objectives and methodologies. Details are as follows.

## A.1  Black-Box-Optimization (BBO) Methods

A straightforward intuition is to view the chip placement task as a black-box-optimization (BBO) problem, where the inner workings of the objective functions are inaccessible, and solutions are evaluated only based on the output metrics.

**Simulated Annealing (SA)** is a heuristic BBO optimization algorithm favored for its simplicity in implementation. Specifically, the SA algorithm generates solutions by perturbing the solution space and then assessing the resulting representation. Different methods have been developed to effectively map representations to placement solutions [7, 32–35], such as sequence pair [36] and B$^*$-tree [37]. Solutions are probabilistically accepted based on an annealing temperature to escape local optima in pursuit of a global optimum. Due to its simplicity in implementation, the SA algorithm often serves as a strong baseline in previous studies. In this work, we incorporate a specific SA implementation [4] utilizing operations like swaps, shifts, and shuffles, and a cost function that balances wirelength, density, and congestion.

**WireMask-EA** [5] is a BBO framework that was recently introduced at the NeurIPS 2023 conference, positioning itself as an innovative approach in the intersection of AI and EDA. The framework utilizes a novel concept called wiremask, which plays a crucial role in guiding the mapping process from genotypes to phenotypes in a greedy manner. The wiremask concept was originally introduced by Lai et al. [14], where it is defined as a matrix that predicts the potential increase in Half-Perimeter Wirelength (HPWL) for each subsequent macro placement on the design canvas. By estimating the wirelength increase, the wiremask helps in making informed decisions during the placement process, thereby potentially improving the quality of the layout. Building upon this concept, Shi et al. [5] extended the framework to integrate several types of Black-Box Optimization (BBO) algorithms, including random search (RS), evolutionary algorithm (EA), and Bayesian optimization (BO), demonstrating the versatility of the approach in handling complex optimization tasks in chip design.

## A.2  Analytical (Gradient-Based) Methods

Analytical methods formulate the optimization objective as an analytical function of module coordinates. This formulation enables efficient solutions through techniques like quadratic programming [38–44] and direct gradient descent [6, 8, 45–48]. This work focuses on the gradient-based algorithms, which are by far the more mainstream algorithms.

**DREAMPlace** [8] is a GPU-accelerated framework that leverages differentiable proxies, such as approximate HPWL, as optimization objectives. It was built upon the previous analytical placement algorithms, ePlace [45] and RePlAce [46], yet significantly speeding up the placement process by using GPUs for acceleration. The series of versions of DREAMPlace introduces diverse differentiable proxies to better align the PPA improvement.

**AutoDMP** [9] extends DREAMPlace by automating hyperparameter tuning through multi-objective Bayesian optimization. It further accelerates the optimization process and reduces manual tuning efforts. At that time, this work showcased the promising potential of integrating GPU-accelerated algorithms with machine learning techniques for automating VLSI design.

## A.3  Reinforcement Learning (RL) Methods

As VLSI systems grow in complexity, RL methods are being explored to enhance placement quality. GraphPlace [10] first models macro placement as a RL problem. Subsequently, DeepPR [12] and PR-Net [13] establish a streamlined pipeline encompassing macro placement, cell placement, and routing. However, they treat density as a soft constraint, which may violate non-overlap constraint during

training. Therefore, in this work, we mainly focus on MaskPlace , ChiPFormer and MaskRegulate, which are recent SOTA algorithms with hard non-overlapping constraints.

**MaskPlace** [14] represents the chip states as pixel-level visual inputs, including a wiremask (recording the HPWL increment for each grid), the viewmask (a global observation of the canvas), and the positionmask (to ensure non-overlapping constraint). Furthermore, it uses dense reward to boost the sample efficiency.

**ChiPFormer** [15] represents the first offline RL method. It is pretrained on various chips via offline RL and then fine-tuned on unseen chips for better efficiency. As a result, the time for placement is significantly reduced.

**MaskRegulate** [25] refines chip placements using RL at the adjustment stage, leveraging precise rewards and incorporating regularity as a key metric. It improves PPA metrics and placement quality across various designs.

## B  Technical details

### B.1  Experimental Details

During the dataset generation process, the macro placement stage is considered part of the floorplan phase. After this phase, the power distribution networks (PDNs) are generated, followed by global placement and detailed placement. During this process, we extracted the DEF files from the macro placement stage, converted them into the required formats based on different algorithmic needs, and executed the corresponding placement algorithms.The macro positions generated by each algorithm were written back into the original DEF files for evaluation. After completing the floorplan stage, we proceeded with global placement and subsequent steps to obtain the final PPA.

For the RL method, we utilized the pre-trained reinforcement learning model provided by the project and leveraged its zero-shot capability to accomplish the task. For ChiPFormer, use the default model and perform 100 online iterations to fine-tune the macro layout.For MaskPlace, run 3000 epochs.Other settings use the defaults. For methods such as DREAMPlace, AutoDMP, and MaskRegulate, their output includes the results of the global placement stage. To ensure consistency in results, we extracted only the macro placement distributions generated by these methods for analysis. In the method of ChipFormer and MaskPlace, the experiments were run on an NVIDIA GeForce RTX 2080 Ti, taking one day for all cases.For the other algorithms, we used 32 CPUs (Intel(R) Xeon(R) CPU E5-2667 v4 @ 3.20GHz), with a total time expenditure of two days.

### B.2  Encountered Errors

Existing open-source EDA toolchains are typically composed of various open-source tools. However, they have certain limitations and flaws in practical use, including functional restrictions and potential bugs. During the design process, we encountered several issues that required resolution. For instance, when using the OpenROAD flow, integrating DEF files back into the original flow caused parsing errors during post-route STA due to special characters (e.g., "/") in some component names, leading to inaccurate analysis results. To address this, we replaced the problematic characters to ensure proper parsing. In addition, OpenROAD lacks robust support for the syntax of SDC files. For example, in the ICCAD2015 dataset, some SDC file content caused read failures, necessitating additional handling during the design flow.

We also identified issues with various open-source placement algorithms during practical application, requiring modifications and optimizations to their code. Specifically, we adjusted the canvas initialization code of MaskPlace and ChipFormer to support non-square die shapes. For ChipFormer, we modified the processes for reading and exporting macro positions. Furthermore, we rewrote the benchmark reading code of MaskRegulate, which was originally limited to the ICCAD2015 dataset, to support our datasets. Through these modifications, we are able to effectively evaluate these algorithms.

# C  Computational Resources

This section summarizes the computational resources. All experiments were executed under the same hardware/software environment to ensure a fair comparison. **Evaluation time** is reported in **minutes** and denotes the wall-clock elapsed time to obtain final PPA after a placement solution is fixed. Concretely, we start the timer when the placement DEF produced by an algorithm is loaded into our backend flow, and stop it after detailed routing and sign-off reports are generated. The time therefore includes global placement, legalization, clock tree synthesis, and routing; it excludes the runtime of the macro/global placer itself, logic synthesis, and any data preparation. Memory usage is reported in **megabytes (MB)**.

Table 4: Evaluation Time (minutes) per Design and Method

| Design | WireMask-EA | SA | AutoDMP | MaskPlace | ChiPFormer | MaskRegulate | DREAMPlace |
|---|---|---|---|---|---|---|---|
| ariane136 | 116 | 117 | 137 | 135 | 126 | 186 | 93 |
| ariane133 | 111 | 141 | 111 | 123 | 131 | 114 | 128 |
| bp | 202 | 212 | 188 | 227 | 211 | 188 | 192 |
| bp_fe | 31 | 29 | 90 | 28 | 354 | 76 | 54 |
| bp_be | 177 | 271 | 396 | 229 | 190 | 63 | 397 |
| bp_multi | 69 | 75 | 100 | 69 | 108 | 84 | 114 |
| swerv_wrapper | 4179 | 4465 | 512 | 4400 | 431 | 183 | 318 |
| vga_lcd | 63 | 51 | 85 | 44 | 51 | 37 | 79 |
| dft68 | 21 | 24 | 13 | 21 | 24 | 17 | 30 |
| or1200 | 76 | 15 | 37 | 17 | 40 | 53 | 47 |
| ethernet | 42 | 27 | 33 | 36 | 29 | 74 | 28 |
| VeriGPU | 38 | 37 | 33 | 37 | 35 | 116 | 38 |
| bp_fe38 | 122 | 157 | 137 | 98 | 48 | 327 | 597 |
| bp_be12 | 87 | 94 | 209 | 156 | 83 | 353 | 317 |
| swerv_wrapper43 | 119 | 252 | 192 | 284 | 183 | 384 | 400 |
| bp_multi57 | 115 | 96 | 202 | 90 | 91 | 221 | 432 |
| ariane81 | 427 | 441 | 508 | 567 | 418 | 221 | 432 |

Table 5: Memory Usage (MB) per Design and Method

| Design | WireMask-EA | SA | AutoDMP | MaskPlace | ChiPFormer | MaskRegulate | DREAMPlace |
|---|---|---|---|---|---|---|---|
| ariane136 | 9007 | 8419 | 8866 | 8926 | 8933 | 9063 | 8603 |
| ariane133 | 8989 | 8516 | 8432 | 8786 | 8594 | 8609 | 8616 |
| bp | 12647 | 11544 | 11595 | 12088 | 12656 | 11750 | 12162 |
| bp_fe | 4053 | 4079 | 4999 | 3888 | 4583 | 4145 | 4016 |
| bp_be | 5353 | 5851 | 5939 | 5791 | 5656 | 4725 | 5759 |
| bp_multi | 7670 | 7712 | 7844 | 7836 | 7940 | 7958 | 7983 |
| swerv_wrapper | 35496 | 12531 | 7269 | 15135 | 7147 | 7448 | 7842 |
| vga_lcd | 5820 | 6112 | 6473 | 6007 | 6190 | 6666 | 6817 |
| dft68 | 3581 | 3441 | 3161 | 3676 | 3350 | 3326 | 3181 |
| or1200 | 3308 | 500 | 3491 | 791 | 790 | 3308 | 3496 |
| ethernet | 3723 | 3630 | 3559 | 3741 | 3575 | 3691 | 3699 |
| VeriGPU | 5386 | 5309 | 5332 | 5344 | 5294 | 5298 | 5251 |
| bp_fe38 | 5993 | 3825 | 3783 | 4403 | 4483 | 3745 | 3971 |
| bp_be12 | 7622 | 5542 | 7138 | 7141 | 5684 | 6736 | 6627 |
| swerv_wrapper43 | 7855 | 7083 | 7060 | 7232 | 7671 | 6989 | 7720 |
| bp_multi57 | 8030 | 7789 | 7152 | 8202 | 9241 | 9233 | 7255 |
| ariane81 | 11472 | 8908 | 10069 | 11433 | 9083 | 10057 | 8718 |

Table 6: The total results from the experiment. MacroHPWL($\mu m$), Regularity($\mu m$), and HPWL($\mu m$) serve as metrics during the placement stage. Congestion (%) and Wirelength($\mu m$) are evaluated in the routing stage. WNS (ns), TNS (ns), NVP, Power (nW), and Area($\mu m^2$) are PPA metrics.

| Benchmark | Method | Intermediate Metrics | | | | | PPA Metrics | | | | |
| | | Placement Metrics | | | Route Metrics | | Timing Performance | | | Power ↓ | Area ↓ |
| | | MacroHPWL ↓ | Regularity ↓ | HPWL ↓ | Congestion ↓ | Wirelength ↓ | WNS ↑ | TNS ↑ | NVP ↓ | | |
| ariane133 | WireMask-EA | 242238 | 114512 | 6433324 | 0.264 | 7651585 | -0.211 | -189.104 | 1943 | 0.351 | 388201 |
| | SA | 199014 | 105329 | 6281435 | 0.251 | 7467599 | -0.543 | -86.418 | **426** | 0.346 | 386449 |
| | AutoDMP | **165788** | 90490 | **6073392** | **0.242** | **7189065** | -0.186 | -132.132 | 1693 | 0.347 | **385105** |
| | MaskPlace | 837745 | 99161 | 8414745 | 0.335 | 9733581 | -0.498 | -766.003 | 2806 | 0.369 | 399260 |
| | ChiPFormer | 228989 | 123006 | 6987169 | 0.282 | 8159913 | -0.49 | -674.917 | 2772 | 0.365 | 391184 |
| | MaskRegulate | 649344 | **42045** | 6938424 | 0.271 | 8087277 | **-0.077** | **-17.829** | 709 | 0.343 | 389803 |
| | DREAMPlace | 248452 | 107875 | 6347837 | 0.261 | 7570321 | -0.374 | -457.43 | 2397 | 0.359 | 386995 |
| | Hier-RTLMP | 568103 | 65804 | 6561610 | 0.266 | 7698115 | -0.129 | -75.056 | 1321 | **0.342** | 389967 |
| ariane136 | WireMask-EA | 253664 | 102336 | 6078412 | 0.251 | **7247980** | -0.177 | -159.48 | 1770 | 0.379 | **393951** |
| | SA | 206782 | 101659 | 6587643 | 0.263 | 7794513 | -0.187 | -169.36 | 1498 | 0.39 | 396525 |
| | AutoDMP | **178085** | 110442 | 6121622 | **0.246** | 7302628 | -0.188 | -166.52 | 1799 | 0.38 | 396119 |
| | MaskPlace | 898193 | 100409 | 8348492 | 0.335 | 9688399 | -0.343 | -382.742 | 2258 | 0.41 | 409283 |
| | ChiPFormer | 301790 | 120650 | 6780158 | 0.278 | 8029029 | -0.282 | -253.365 | 2013 | 0.396 | 397091 |
| | MaskRegulate | 616012 | **44090** | 6908821 | 0.272 | 8089823 | -0.194 | -154.2 | 1766 | **0.378** | 397739 |
| | DREAMPlace | 271713 | 106918 | **6075063** | 0.247 | 7305799 | -0.259 | -247.105 | 2001 | 0.387 | 394257 |
| | Hier-RTLMP | 525799 | 74380 | 6392817 | 0.262 | 7582918 | **-0.116** | **-50.597** | 1079 | 0.379 | 397302 |
| bp | WireMask-EA | 23159 | 17726 | 8936778 | 0.461 | 10430310 | -4.818 | -72.26 | 1367 | 0.508 | 533310 |
| | SA | 28700 | 15631 | 8804023 | 0.442 | 10290194 | -4.451 | -74.026 | 1719 | 0.501 | 534082 |
| | AutoDMP | 30637 | 18123 | 8716770 | 0.44 | 10247761 | -4.861 | -211.311 | 2627 | 0.498 | 536293 |
| | MaskPlace | 62427 | 17401 | 9081986 | 0.455 | 10599862 | -4.741 | -1316.7 | 6150 | 0.501 | 532615 |
| | ChiPFormer | 23948 | 18854 | 8787969 | 0.466 | 10556917 | -4.761 | -263.491 | 955 | 0.5 | 532571 |
| | MaskRegulate | 46161 | **8008** | 8048760 | 0.418 | 9475677 | **-4.371** | **-35.503** | 532 | 0.505 | 528872 |
| | DREAMPlace | 30257 | 16972 | 8434472 | 0.433 | 9811774 | -4.524 | -156.102 | 1670 | 0.495 | 528154 |
| | Hier-RTLMP | **22621** | 14169 | **8010716** | **0.414** | **9379330** | -4.822 | -22458.4 | 14758 | **0.488** | **525713** |
| bp_be | WireMask-EA | **12132** | 4401 | 3337281 | 0.617 | 4192684 | -0.603 | -48.936 | 111 | 0.147 | 123172 |
| | SA | 14069 | 4008 | 3326938 | 0.633 | 4317839 | -0.937 | -79.232 | 111 | 0.151 | 122659 |
| | AutoDMP | 16220 | 4496 | 3720895 | 0.7 | 4938884 | -0.898 | -70.97 | 220 | 0.157 | 126114 |
| | MaskPlace | 20794 | 4140 | 3422765 | 0.648 | 4434133 | -0.669 | -45.159 | 111 | 0.147 | 123782 |
| | ChiPFormer | 12226 | 5349 | 3257074 | 0.61 | 4145308 | -0.836 | -67.798 | 111 | 0.149 | 123200 |
| | MaskRegulate | 16338 | **2809** | 3121381 | 0.519 | 3684330 | **-0.495** | -39.754 | 110 | 0.142 | 121308 |
| | DREAMPlace | 17167 | 4942 | 3388866 | 0.628 | 4289216 | -0.784 | -45.56 | 111 | 0.149 | 124045 |
| | Hier-RTLMP | 13575 | 4333 | 3146056 | 0.591 | 4028731 | -0.529 | **-39.002** | 110 | 0.146 | 122135 |
| bp_fe | WireMask-EA | 45868 | 5343 | 2617101 | 0.579 | 3181490 | -0.332 | -67.505 | 539 | 0.179 | 74335 |
| | SA | 41649 | 3517 | 2409585 | 0.546 | 3010920 | -0.314 | -12.508 | 139 | 0.176 | 72022 |
| | AutoDMP | **40442** | 4281 | 2423670 | 0.684 | 3610750 | **-0.092** | -1.657 | 43 | 0.168 | 70595 |
| | MaskPlace | 65234 | 3893 | 2261692 | 0.521 | 2867419 | -0.168 | -2.73 | 49 | 0.17 | 72112 |
| | ChiPFormer | 65234 | 3893 | 2261692 | 0.595 | 3208864 | -0.752 | -33.906 | 257 | 0.173 | 72112 |
| | MaskRegulate | 45537 | **3043** | 2281612 | 0.55 | 3021127 | -0.232 | -13.622 | 113 | **0.167** | 70687 |
| | DREAMPlace | 50669 | 4377 | 2125704 | 0.528 | 2871636 | -0.528 | -38.339 | 159 | 0.168 | **70523** |
| | Hier-RTLMP | 48682 | 4083 | **2100584** | 0.481 | **2628466** | -0.116 | **-1.007** | 110 | 0.167 | 70658 |
| bp_multi | WireMask-EA | 30929 | 15190 | 5367993 | 0.42 | 6124013 | -5.843 | -4063.81 | 10404 | 0.545 | 268133 |
| | SA | 36422 | 13761 | 5127229 | 0.387 | 5849303 | -5.479 | -3280.39 | 9967 | 0.538 | 267251 |
| | AutoDMP | 36517 | 17625 | 5554418 | 0.41 | 6210775 | -5.623 | -3492.19 | 9454 | 0.539 | 265453 |
| | MaskPlace | 134103 | 15149 | 5580849 | 0.437 | 6384545 | **-5.436** | -4111.42 | 9676 | 0.541 | 269016 |
| | ChiPFormer | 28916 | 17044 | 5202164 | 0.407 | 5938064 | -5.605 | -4453.74 | 9779 | 0.544 | 266880 |
| | MaskRegulate | 44104 | **8347** | 4651180 | **0.37** | **5398697** | -5.453 | -2991.5 | **8089** | **0.534** | **260721** |
| | DREAMPlace | 39400 | 16449 | 5214381 | 0.409 | 5971959 | -5.622 | -3192.49 | 10119 | 0.536 | 267504 |
| | Hier-RTLMP | **22785** | 11794 | 4724466 | 0.372 | 5435399 | -5.706 | **-2926.31** | 9867 | 0.535 | 262623 |
| swerv_wrapper | WireMask-EA | 92304 | 16087 | 5052232 | 0.445 | 6203022 | -1.12 | -1052.14 | 1791 | 0.296 | 235525 |
| | SA | 108068 | 15756 | 4637819 | 0.383 | 5561268 | -1.033 | -863.393 | 1485 | 0.273 | 230076 |
| | AutoDMP | 101651 | 18086 | 4214108 | 0.356 | 5173002 | -0.941 | -903.64 | 1478 | 0.27 | 229290 |
| | MaskPlace | 282636 | 14743 | 4634862 | 0.378 | 5484915 | -0.768 | -582.361 | **1363** | 0.271 | 230706 |
| | ChiPFormer | **89998** | 17512 | 4718772 | 0.408 | 5685641 | -1.352 | -1496.87 | 1537 | 0.277 | 233285 |
| | MaskRegulate | 221155 | **11265** | 3991734 | **0.325** | 4731291 | -0.67 | **-516.377** | 1365 | 0.266 | 228183 |
| | DREAMPlace | 105719 | 15149 | 3965871 | 0.338 | 4730011 | -0.744 | -572.391 | 1415 | 0.266 | 228845 |
| | Hier-RTLMP | 118198 | 16732 | **3804541** | 0.326 | **4550107** | **-0.66** | -613.774 | 1435 | **0.265** | 226536 |
| dft68 | WireMask-EA | 221950 | 39095 | 1346077 | 0.112 | 1505776 | -0.335 | -64.921 | 278 | 0.234 | 87852 |
| | SA | 246786 | 36998 | 1327568 | 0.106 | 1476277 | -0.347 | -63.158 | 276 | **0.226** | 87679 |
| | AutoDMP | **190255** | 42553 | **1075304** | **0.087** | **1220031** | -0.293 | -58.685 | 276 | **0.223** | 87624 |
| | MaskPlace | 752034 | 37485 | 2408871 | 0.192 | 2577200 | -0.293 | -60.844 | 278 | 0.247 | 92363 |
| | ChiPFormer | 376737 | 42956 | 1452266 | 0.114 | 1596730 | -0.301 | -61.864 | 276 | 0.234 | 87462 |
| | MaskRegulate | 336667 | **29892** | 1509324 | 0.116 | 1628526 | -0.31 | -64.544 | 285 | 0.231 | **85163** |
| | DREAMPlace | 198013 | 46027 | 1150751 | 0.092 | 1292984 | -0.31 | -62.484 | 277 | 0.225 | 86322 |
| | Hier-RTLMP | 513984 | 34462 | 1751899 | 0.135 | 1888167 | **-0.292** | **-57.656** | **275** | 0.237 | 88779 |

# D More Results

All the results from the experiment are in Tables 6-7.

Table 7: Continuation of the total results from the experiment. MacroHPWL($\mu m$), Regularity($\mu m$), and HPWL($\mu m$) serve as metrics during the placement stage. Congestion (%) and Wirelength($\mu m$) are evaluated in the routing stage. WNS (ns), TNS (ns), NVP, Power (nW), and Area($\mu m^2$) are PPA metrics.

| Benchmark | Method | Intermediate Metrics | | | | | PPA Metrics | | | | |
| | | Placement Metrics | | | Route Metrics | | Timing Performance | | | Power ↓ | Area ↓ |
| | | MacroHPWL ↓ | Regularity ↓ | HPWL ↓ | Congestion ↓ | Wirelength ↓ | WNS ↑ | TNS ↑ | NVP ↓ | | |
|---|---|---|---|---|---|---|---|---|---|---|---|
| ethernet | WireMask-EA | 33333 | 23562 | 1039116 | 0.356 | 1339544 | -0.225 | -11.497 | 181 | 0.121 | 98142 |
| | SA | 32535 | 23021 | 907604 | 0.303 | 1155201 | -0.151 | -4.9 | 87 | 0.119 | 96782 |
| | AutoDMP | 27806 | 17475 | 862110 | **0.285** | **1086531** | -0.161 | **-2.691** | **81** | **0.118** | 96593 |
| | MaskPlace | 44773 | 18065 | 941947 | 0.32 | 1205506 | -0.188 | -5.061 | 105 | 0.12 | 97243 |
| | ChiPFormer | 34756 | 18018 | 905432 | 0.299 | 1137392 | -0.178 | -5.728 | 91 | 0.118 | 96908 |
| | MaskRegulate | 38545 | **8979** | 872538 | 0.29 | 1098309 | -0.311 | -8.686 | 86 | 0.118 | **96087** |
| | DREAMPlace | **27407** | 15645 | 876320 | 0.298 | 1117049 | **-0.121** | -3.548 | 90 | 0.119 | 96925 |
| | Hier-RTLMP | 27959 | 13597 | **859203** | 0.286 | 1092044 | -0.199 | -10.524 | 176 | 0.118 | 96157 |
| vga_lcd | WireMask-EA | 63144 | 47858 | 1651480 | **0.122** | **2403183** | -1.419 | -525.233 | 3306 | 0.188 | 250788 |
| | SA | 62127 | 47979 | 1617172 | 0.123 | 2423446 | -2.525 | -2513.52 | 6231 | 0.187 | 251728 |
| | AutoDMP | **50597** | 54164 | 1579406 | 0.127 | 2577725 | -1.677 | -5270.93 | 14440 | 0.192 | 263853 |
| | MaskPlace | 116002 | 43041 | 1857369 | 0.136 | 2676714 | -1.451 | -209.683 | **2720** | 0.191 | 253809 |
| | ChiPFormer | 55335 | 46957 | 1573687 | 0.128 | 2516897 | -1.502 | -7975.91 | 18370 | 0.193 | 254257 |
| | MaskRegulate | 86314 | **28023** | 1619866 | 0.136 | 2765299 | -1.805 | -2229.64 | 9010 | **0.185** | **163340** |
| | DREAMPlace | 70247 | 43622 | **1481247** | 0.125 | 2465370 | **-1.12** | -1458.73 | 7313 | 0.189 | 263523 |
| | Hier-RTLMP | 101530 | 28761 | 1716574 | 0.133 | 2706057 | -1.214 | -676.131 | 6430 | 0.191 | 260105 |
| VeriGPU | WireMask-EA | 2134 | 6930 | **1111197** | **0.179** | 1587238 | -0.462 | -63.542 | 656 | 0.096 | 152631 |
| | SA | 2070 | 6480 | 1133066 | 0.182 | 1613611 | -0.411 | -101.937 | 1071 | 0.096 | 152835 |
| | AutoDMP | 2095 | 5778 | 1131313 | 0.183 | 1623046 | -0.508 | -70.665 | **439** | 0.095 | 152990 |
| | MaskPlace | 2856 | 5192 | 1173636 | 0.188 | 1661907 | -0.623 | -150.034 | 947 | **0.093** | **152175** |
| | ChiPFormer | **1587** | 4439 | 1143320 | 0.183 | 1617706 | -0.375 | -61.958 | 641 | 0.095 | 152899 |
| | MaskRegulate | 2772 | **4057** | 1214396 | 0.188 | 1710181 | -0.436 | -75.662 | 502 | 0.093 | 152338 |
| | DREAMPlace | 2297 | 6092 | 1192232 | 0.19 | 1687203 | **-0.291** | -80.186 | 630 | 0.096 | 153476 |
| | Hier-RTLMP | 2364 | 5022 | 1196418 | 0.189 | 1670147 | -0.294 | **-51.109** | 491 | 0.094 | 153084 |
| ariane81 | WireMask-EA | 208944 | 66638 | 6263567 | 0.381 | 8235930 | -0.813 | -1942.48 | 3312 | 0.197 | 344495 |
| | SA | 199555 | 54578 | 5942475 | 0.332 | 7374859 | -0.325 | -404.722 | 2143 | 0.193 | 341581 |
| | AutoDMP | 162147 | 66033 | 5266932 | 0.288 | 6405132 | -0.147 | **-101.844** | **1402** | 0.188 | 339448 |
| | MaskPlace | 739756 | 55947 | 7069029 | 0.423 | 9163711 | -0.692 | -1438.37 | 3231 | 0.203 | 350442 |
| | ChiPFormer | 185541 | 63300 | 5742238 | 0.318 | 6844088 | -0.184 | -109.88 | 1449 | 0.193 | 343524 |
| | MaskRegulate | 436868 | **38070** | 5820626 | 0.326 | 7255339 | -0.148 | -103.496 | 1455 | 0.191 | 340730 |
| | DREAMPlace | **156297** | 61184 | **4559488** | **0.254** | **5643862** | **-0.143** | -116.533 | 1657 | **0.186** | **335894** |
| | Hier-RTLMP | 330885 | 53896 | 5538206 | 0.323 | 6980976 | -0.298 | -397.887 | 2151 | 0.189 | 339853 |
| bp_be12 | WireMask-EA | 801604 | 6488 | 3693994 | 0.558 | 4512177 | -1.215 | -160.407 | 343 | 0.076 | 91574 |
| | SA | 782211 | 5837 | 3755117 | 0.543 | 4304811 | **-0.75** | -70.518 | 114 | 0.076 | 94456 |
| | AutoDMP | 722030 | 6034 | 3504039 | 0.51 | 4261752 | -0.944 | -90.002 | 129 | **0.074** | 87899 |
| | MaskPlace | 775553 | 5497 | 3699082 | 0.523 | 4365827 | -1.051 | -113.241 | 308 | 0.075 | 88779 |
| | ChiPFormer | **719018** | 5912 | 3800124 | 0.549 | 4366388 | -0.835 | **-66.482** | 114 | 0.075 | 93953 |
| | MaskRegulate | 848084 | **3480** | 3781833 | 0.537 | 4465025 | -0.758 | -72.157 | 116 | 0.075 | 90972 |
| | DREAMPlace | 731770 | 6381 | **3215579** | **0.471** | **3797049** | -0.775 | -74.772 | 114 | 0.074 | **86949** |
| | Hier-RTLMP | 759741 | 4758 | 3528859 | 0.534 | 4297826 | -0.89 | -87.346 | 115 | 0.075 | 88026 |
| bp_fe38 | WireMask-EA | 928352 | 18728 | 3404465 | 0.475 | 4003637 | -1.486 | -676.669 | 1068 | 0.112 | 62662 |
| | SA | 892633 | 14572 | **2935496** | 0.389 | 3237419 | -1.453 | **-334.63** | 587 | **0.111** | **59398** |
| | AutoDMP | **863228** | 17699 | 2957685 | 0.395 | 3292141 | -1.403 | -526.071 | 781 | 0.111 | 59751 |
| | MaskPlace | 1074969 | 17056 | 3204116 | 0.454 | 3652153 | -1.358 | -494.692 | 738 | 0.111 | 60497 |
| | ChiPFormer | 874067 | 18921 | 3899590 | 0.552 | 4446133 | -2.015 | -1896.47 | 1864 | 0.112 | 67309 |
| | MaskRegulate | 986798 | **11323** | 3019194 | 0.398 | 3309468 | **-1.288** | -374.213 | 623 | 0.111 | 59552 |
| | DREAMPlace | 887412 | 19321 | 3213779 | 0.435 | 3626122 | -1.351 | -376.483 | 613 | 0.111 | 61727 |
| | Hier-RTLMP | 960487 | 19504 | 3289998 | 0.463 | 3733979 | -1.998 | -808.689 | 1383 | 0.111 | 62706 |
| bp_multi57 | WireMask-EA | 1134707 | 37830 | 7065213 | 0.575 | 7825719 | -3.064 | -887.269 | 566 | 0.107 | 212841 |
| | SA | 1114175 | 32659 | 7598877 | 0.587 | 8363004 | **-1.962** | -418.758 | 353 | 0.11 | 216864 |
| | AutoDMP | **499382** | 38144 | 5956153 | 0.474 | 6764802 | -2.474 | -622.456 | 412 | 0.107 | 205661 |
| | MaskPlace | 1128354 | 32415 | 8234160 | 0.633 | 9012536 | -2.043 | -461.315 | 365 | 0.11 | 220663 |
| | ChiPFormer | 1167727 | 38451 | 7753888 | 0.633 | 8676775 | -2.971 | -922.214 | 689 | 0.107 | 218603 |
| | MaskRegulate | 887050 | **23312** | 8112323 | 0.616 | 8809855 | -3.858 | -1225.7 | 551 | 0.107 | 213761 |
| | DREAMPlace | 515292 | 36648 | 5835798 | **0.46** | 6558793 | -1.967 | **-399.415** | 349 | **0.105** | 205736 |
| | Hier-RTLMP | 739354 | 25468 | **5670985** | 0.467 | **6333021** | -2.499 | -621.463 | 407 | 0.105 | **203747** |
| swerv_wrapper43 | WireMask-EA | 170470 | 32345 | 5445062 | 0.286 | 6689880 | -0.624 | -465.246 | 1073 | 0.262 | 237381 |
| | SA | 159134 | 24460 | 4422281 | 0.216 | 5193274 | -0.555 | -424.13 | 1312 | 0.255 | 228015 |
| | AutoDMP | **134513** | 30710 | 4074313 | **0.198** | **4777988** | -0.656 | -487.128 | 1262 | 0.255 | 226496 |
| | MaskPlace | 553676 | 28096 | 5995283 | 0.293 | 6852646 | -0.712 | -545.3 | 1508 | 0.276 | 240865 |
| | ChiPFormer | 169043 | 32229 | 4361625 | 0.219 | 5135204 | -0.638 | -487.583 | 1228 | 0.256 | 228100 |
| | MaskRegulate | 161152 | **14528** | 4332287 | 0.212 | 5103448 | **-0.457** | **-310.164** | 954 | **0.254** | 227237 |
| | DREAMPlace | 155826 | 33516 | **4046946** | 0.205 | 4814504 | -0.568 | -430.422 | 1136 | 0.26 | 226821 |
| | Hier-RTLMP | 198089 | 23576 | 4194027 | 0.211 | 4949116 | -0.482 | -352.959 | 1222 | 0.256 | **225634** |

# E   Additional Experiments and Analysis

## E.1   Commercial Baselines and Cross-Verification

To contextualize academic methods against mature commercial EDA tools and to verify the fidelity of our open-source evaluation flow, we conduct two complementary studies.

**Commercial baselines.** We run Synopsys' commercial placer on six representative designs from our dataset and then evaluate the resulting layouts with our open-source flow to obtain final PPA. The absolute results and their normalization against our baseline are reported in Table 8 and Table 9, respectively. Overall, the commercial tool shows advantages on several timing- and violation-related metrics, underscoring the headroom that remains for open-source and AI-based approaches, particularly in PPA optimization.

Table 8: Evaluation results using a commercial EDA tool (Synopsys).

| | MacroHPWL ($\mu$m) | Regularity ($\mu$m) | HPWL ($\mu$m) | Cong. | Wirelength ($\mu$m) | WNS (ns) | TNS (ns) | NVP | Power (nW) | Area ($\mu$m$^2$) |
|---|---|---|---|---|---|---|---|---|---|---|
| ariane133 | 180241 | 118308 | 6077457.9 | 0.244 | 7264950 | -0.0520 | -12.34 | 402 | 0.304 | 387311 |
| ariane136 | 186767 | 117957 | 6130882 | 0.248 | 7355342 | -0.0686 | -21.34 | 591 | 0.325 | 396284 |
| bp | 23048 | 17401 | 8401960 | 0.515 | 944277 | -4.12 | -72.34 | 621 | 0.496 | 534217 |
| bp_be | 18821 | 3681 | 3346075 | 0.607 | 4150867 | -0.849 | -41.63 | 144 | 0.123 | 123221 |
| bp_fe | 36544 | 4110 | 2356003 | 0.583 | 3161588 | -0.0562 | -0.7112 | 28 | 0.142 | 72962 |
| swerv_wrapper | 96692 | 12794 | 4143879 | 0.355 | 4970401 | -0.703 | -697.8 | 1500 | 0.240 | 229277 |

Table 9: Normalized results of the commercial tool relative to the open-source baseline.

| | MacroHPWL | Regularity | HPWL | Cong. | Wirelength | WNS | TNS | NVP | Power | Area |
|---|---|---|---|---|---|---|---|---|---|---|
| ariane133 | 0.317 | 1.798 | 0.926 | 0.919 | 0.944 | 0.404 | 0.164 | 0.304 | 0.889 | 0.993 |
| ariane136 | 0.355 | 1.586 | 0.959 | 0.947 | 0.970 | 0.592 | 0.422 | 0.548 | 0.859 | 0.997 |
| bp | 1.019 | 1.228 | 1.049 | 1.232 | 0.101 | 0.855 | 0.003 | 0.042 | 1.016 | 1.016 |
| bp_be | 1.386 | 0.850 | 1.064 | 1.028 | 1.030 | 1.606 | 1.067 | 1.309 | 0.846 | 1.009 |
| bp_fe | 0.751 | 1.007 | 1.122 | 1.214 | 1.203 | 0.483 | 0.706 | 0.875 | 0.854 | 1.033 |
| swerv_wrapper | 0.818 | 0.765 | 1.089 | 1.091 | 1.092 | 1.066 | 1.137 | 1.045 | 0.906 | 1.012 |
| Avg. | 0.774 | 1.205 | 1.035 | 1.072 | 0.890 | 0.834 | 0.583 | 0.687 | 0.895 | 1.010 |

**Cross-verification with a commercial flow.** To assess evaluation consistency, we take placements produced by multiple AI algorithms and evaluate them twice: once with the commercial sign-off flow and once with our open-source flow. The comparison in Table 10 shows that the relative ranking of algorithms is preserved across the two flows. This consistency demonstrates that ChiPBench is a reliable and trustworthy framework for differentiating the quality of placement algorithms, which is the core purpose of our benchmark.

Table 10: Comparison of Results Between Commercial EDA Tool Flow and Our Proposed Flow (after Normalized).

| Commercial EDA Tool | | | | | Our Open-Source Flow | | | | |
|---|---|---|---|---|---|---|---|---|---|
| Algorithm | WNS | TNS | NVP | Power | Algorithm | WNS | TNS | NVP | Power |
| AutoDMP | 1.129 | 1.041 | 1.271 | 1.647 | AutoDMP | 1.316 | 1.501 | 1.120 | 1.022 |
| WireMask-EA | 1.161 | 1.057 | 1.014 | 1.018 | WireMask-EA | 1.280 | 1.580 | 1.039 | 1.033 |
| ChiPFormer | 1.132 | 1.057 | 1.014 | 1.018 | ChiPFormer | 1.976 | 3.422 | 1.243 | 1.035 |
| DREAMPlace | 1.331 | 1.483 | 1.276 | 1.119 | DREAMPlace | 1.560 | 2.259 | 1.037 | 1.018 |
| MaskPlace | 1.355 | 1.151 | 1.017 | 1.030 | MaskPlace | 1.818 | 3.425 | 1.188 | 1.036 |
| **MaskRegulate** | **0.961** | **0.813** | **0.976** | 1.050 | **MaskRegulate** | **0.962** | **0.935** | **0.769** | 1.002 |
| SA | 1.291 | 1.158 | 0.984 | 1.030 | SA | 1.801 | 1.392 | 0.716 | 1.022 |
| Hier-RTLMP | 1.000 | 1.000 | 1.000 | **1.000** | Hier-RTLMP | 1.000 | 1.000 | 1.000 | **1.000** |

## E.2 Evaluation of Other Stages

Beyond placement, our dataset and workflow are stage-agnostic and can be used to benchmark algorithms at other points in the EDA flow. In this section, we evaluate logic synthesis. Specifically, we compare the widely used open-source synthesizer Yosys with a commercial tool (Synopsys Design Compiler, DC) [49], on our dataset.

For each design, we synthesize the RTL with either Yosys or DC under identical technology libraries and timing constraints. The resulting gate-level netlists are then fed into the same open-source backend to ensure fairness—running floorplanning, placement, clock-tree synthesis, and detailed routing—after which we report the final PPA metrics. The results are summarized in Table 11.

From Table 11, the comparison reveals that there is still a significant performance gap between existing open-source synthesis tools and mature commercial solutions. We believe that the proposed evaluation framework and dataset can play a vital role in accelerating the advancement of open-source synthesis tools.

Table 11: Comparison of Results between a commercial tool (Synopsys DC) and open-source tool Yosys. The metrics are WNS (ns), TNS (ns), NVP, Power (nW), and Area($\mu m^2$)

| | WNS (Yosys) | WNS (DC) | TNS (Yosys) | TNS (DC) | NVP (Yosys) | NVP (DC) | Power (Yosys) | Power (DC) | Area (Yosys) | Area (DC) |
|---|---|---|---|---|---|---|---|---|---|---|
| ariane133 | -0.129 | -0.041 | -75.056 | -41.344 | 1321 | 452 | 0.342 | 0.317 | 389967 | 386121 |
| ariane136 | -0.116 | -0.056 | -50.597 | -34.421 | 1079 | 718 | 0.379 | 0.325 | 397302 | 375192 |
| bp | -4.822 | -4.011 | -22458.4 | -95.1250 | 14758 | 511 | 0.488 | 0.492 | 525713 | 535112 |
| bp_be | -0.529 | -0.497 | -39.002 | -32.510 | 110 | 151 | 0.416 | 0.395 | 122135 | 121221 |
| bp_fe | -0.116 | -0.062 | -1.007 | -0.625 | 32 | 26 | 0.167 | 0.151 | 70658 | 715526 |
| swerv_wrapper | -0.660 | -0.813 | -613.774 | -717.243 | 1435 | 1612 | 0.265 | 0.241 | 226536 | 221256 |

# F   License

We include the following licenses for the code and raw data we used in this paper.

- Yosys:ISC
- OpenROAD:BSD-3-Clause
- ariane133:SOLDERPAD HARDWARE
- ariane136:SOLDERPAD HARDWARE
- bp:BSD-3-Clause
- bp_be:BSD-3-Clause
- bp_fe:BSD-3-Clause
- bp_multi:BSD-3-Clause
- swerv_wrapper:Apache

