# OpenReview forum: "Benchmarking End-To-End Performance of AI-Based Chip Placement Algorithms"
_NeurIPS.cc/2025/Datasets_and_Benchmarks_Track — NeurIPS 2025 Datasets and Benchmarks Track poster_

### Official Review · Reviewer_9LHR · 2025-06-11

**Rating:** 4
**Confidence:** 4

**Summary:**

This paper presents a valuable contribution to the EDA community by offering a unified evaluation framework for AI-based macro placement methods. By integrating macro placement results into the OpenROAD open-source EDA flow, the authors provide a practical pipeline for assessing AI algorithms under realistic design constraints. This integration significantly lowers the barrier for researchers to benchmark their methods, and it promotes a more rigorous standard for comparing the effectiveness of different approaches.

One of the notable strengths of the work lies in its methodological design. The evaluation flow is comprehensive, incorporating key physical design metrics such as congestion, wirelength, timing slack, and violations. These are all critical indicators of placement quality, and the authors have made a reasonable effort to ensure the benchmarking process reflects real-world physical design requirements. Although most of the testcases are derived from the OpenROAD project, the authors have extended the dataset by including more designs, which enhances the robustness of their evaluation and reflects a substantial implementation effort.

However, while the motivation behind unifying the evaluation process is appreciated, the paper overstates the scope of its contribution by positioning it as benchmarking for "AI-based chip placement." In reality, the focus is limited to macro placement, a relatively small and manageable subtask compared to the much more complex global standard cell placement. This distinction is important and should have been emphasized more clearly to avoid giving a misleading impression of the framework's (and also the AI-based placement community's) capabilities. In addition, it would be better to open-source the implementation of tested methods and test more recent works.

From a writing and presentation perspective, the paper could benefit from some restructuring. A large portion of the content is devoted to introductory background, which could be reduced in favor of a more detailed discussion on testcase selection, metric definitions, and experimental insights. The explanation of the evaluation metrics could also be improved by providing clearer, more structured descriptions, possibly with supporting figures or formulas to aid understanding. Additionally, the citation formatting is inconsistent, alternating between [Names, Year] and Names [Year], which disrupts the flow of the text and suggests a need for more careful proofreading. Moreover, It would be better if the authors could cite the original papers/repos of the testcases.

In summary, this paper makes a meaningful and timely contribution by providing an accessible and standardized framework for evaluating AI-based macro placement algorithms. While the claims about chip-level placement are overstated, the work is well-executed and fills an important gap in the research ecosystem. With some refinement in scope definition, writing, and presentation, this paper has the potential to become a cornerstone reference for future work in AI-assisted physical design.

**Dataset Code Accessibility:**

Partly

**Dataset Code Comments:**

The data and evaluation code are accessible. However, it would be better to include the code for reproducing the experiments.

**Ethical Considerations:**

No, there are no or only very minor ethics concerns

**Final Justification:**

This paper presents a valuable contribution by introducing a standardized evaluation framework for AI-based macro placement, integrated within the OpenROAD flow. The methodology is thoughtfully designed, and the benchmarking metrics reflect key physical design objectives. While the scope of the contribution is somewhat narrower than claimed, the framework still fills an important gap in the research community. The authors' rebuttal addresses the reviewer’s concerns to some extent, though certain issues remain, such as the reliance on existing OpenROAD testcases and minor inconsistencies in runtime results. Despite these limitations, the implementation effort is commendable, and the work has the potential to serve as a foundational reference for future studies in AI-driven physical design.

Thus, the reviewer believes that this paper is suitable for borderline acceptance.

**Limitations Weaknesses:**

Despite its practical contributions, the work also exhibits several important limitations that temper its overall impact. The paper presents itself as a framework for evaluating "AI-based chip placement," yet in practice, it focuses exclusively on macro placement, which constitutes only a small and relatively simple subproblem within the broader placement task. This overstatement of scope may mislead readers about the actual capabilities of the framework and limits its significance in addressing more complex and large-scale placement challenges, such as global standard-cell placement. The methods and metrics used—such as wirelength, congestion, and timing slack—are standard in physical design. However, the review considers that the efficiency and scalability of macro placement algorithms should also be considered. Scalability is a real problem in this field. For example, related work [*1] avoids using large-scale testcases that take a long time in evaluation, but traditional placement algorithms like [*2] are super fast. Finally, the writing and organization detract from the work's polish and clarity. Excessive background coverage, inconsistent citation formatting, and limited detail on the metric computation and testcase selection reduce the paper's effectiveness in communicating its core contributions.

[*1] Yunqi Shi, Ke Xue, Lei Song, Chao Qian, "Macro Placement by Wire-Mask-Guided Black-Box Optimization", NeurIPS 2023.
[*2] Yibo Lin, Shounak Dhar, Wuxi Li, Haoxing Ren, Brucek Khailany, David Z. Pan, "DREAMPlace: Deep Learning Toolkit-Enabled GPU Acceleration for Modern VLSI Placement", DAC 2019.

**Strengths Contributions:**

The key strength of this work lies in its timely and practical contribution to the emerging field of AI-driven electronic design automation. While numerous studies propose AI models for macro placement, this paper stands out by addressing the critical gap in evaluation infrastructure. Its central contribution is the development of an end-to-end benchmarking framework that integrates AI-generated macro placements into the OpenROAD full physical design flow, enabling consistent, reproducible, and meaningful assessments of placement quality. This is a significant step forward in unifying evaluation standards in a field that has previously lacked common benchmarks and comparable metrics. The novelty of the work is not in proposing a new placement algorithm, but in its systematization of the evaluation process, including comprehensive consideration of physical design metrics such as congestion, wirelength, timing slack, and violations. This infrastructure has the potential to become a foundational tool for future research, fostering more rigorous and fair comparisons among AI-based approaches. Furthermore, the work is highly relevant given the growing interest in applying machine learning to chip design, and its integration with OpenROAD ensures accessibility and practical adoption by both academia and industry. By focusing on evaluation, the paper complements algorithmic innovations and supports the broader goal of reproducible and transparent progress in AI-assisted physical design.

---

> ### Author Rebuttal · Authors · 2025-07-31
>
> Dear reviewer 9LHR,
>
> Thank you for your insightful and valuable comments. We sincerely hope our rebuttal could adequately address your concerns. If so, we would deeply appreciate it if you could consider raising your score. If not, please let us know your further concerns, and we will continue actively responding to your comments.
>
> # Weakness 1
>
> > This overstatement of scope may mislead readers about the actual capabilities of the framework and limits its significance in addressing more complex and large-scale placement challenges, such as global standard-cell placement.
>
> Thank you for this very insightful comment.
> We acknowledge that our paper's experiments primarily focused on macro placement, which may have obscured the framework's broader utility. To be clear, the **ChiPBench evaluation pipeline is stage-agnostic and is fully capable of evaluating global placers.** Our experimental focus was motivated by the recent high volume of AI research specifically in macro placement.
>
> To prove this capability, we have now benchmarked the well-known global placer, **DREAMPlace**, within our framework. The table below shows its final PPA results compared to the OpenROAD baseline.
>
> **Table1:Evaluation Results**
>
> | OpenROAD      |        |          |       |       |        | DREAMPlace    |          |       |      |         |         |
> | ------------- | ------ | -------- | ----- | ----- | ------ | ------------- | -------- | ----- | ---- | ------- | ------- |
> | Algorithm     | WNS    | TNS      | NVP   | Power | Area   | Algorithm     | WNS      | TNS   | NVP  | Power   | Area    |
> | ariane133     | -0.129 | -75.056  | 1321  | 0.342 | 389967 | ariane133     | -0.20801 | -203  | 1985 | 0.34192 | 386932  |
> | ariane136     | -0.116 | -50.597  | 1079  | 0.379 | 397302 | ariane136     | -0.32328 | -430  | 2222 | 0.38138 | 395266  |
> | bp            | -4.822 | -22458.4 | 14758 | 0.488 | 525713 | bp            | -4.82547 | -981  | 4790 | 0.49053 | 520400  |
> | bp_be         | -0.529 | -39.002  | 110   | 0.146 | 122135 | bp_be         | -0.1676  | -16.1 | 220  | 0.14536 | 122721  |
> | bp_fe         | -0.116 | -1.007   | 32    | 0.167 | 70658  | bp_fe         | -1.29256 | -18.2 | 107  | 0.16556 | 66190.9 |
> | swerv_wrapper | -0.66  | -613.774 | 1435  | 0.265 | 226536 | swerv_wrapper | -0.69256 | -640  | 1446 | 0.26774 | 226462  |
>
>
>
>
> # Weakness 2
>
> > However, the review considers that the efficiency and scalability of macro placement algorithms should also be considered. Scalability is a real problem in this field.
>
> Thank you for these important suggestions. We agree that these aspects are critical for a comprehensive benchmark.
>
> We acknowledge that runtime is a critical factor, especially when comparing against fast traditional methods. In our submitted paper, we already provide detailed data on evaluation time (Table 4) and memory usage (Table 5) for all six algorithms across all 20 designs. However, we did not elaborate on this in the main text. We will add a dedicated subsection to our analysis (Section 6) to discuss these results, explicitly addressing the trade-offs between the PPA and the computational cost of different AI-based approaches. This will directly address your concern regarding efficiency and scalability.
>
>
> # Weakness 3
>
> > Excessive background coverage, inconsistent citation formatting, and limited detail on the metric computation and testcase selection reduce the paper's effectiveness in communicating its core contributions.
>
> Thank you for your careful reading. We will fix the errors.

---

> > ### Comment · Reviewer_9LHR · 2025-08-05
> >
> > Thank you to the authors for their feedback. The responses address the reviewer’s questions to some extent.
> >
> > Regarding the answer to weakness 1, the reviewer appreciates the additional experiments on global placement. However, these testcases have been provided by the OpenROAD-flow-script project. Thus, the reviewer considers the contribution to be limited.
> >
> > For the answer to Weakness 2, the reviewer appreciates the inclusion of runtime information in Table 4. However, some results appear inconsistent with the reviewer’s experience. For instance, macro placement typically involves a small number of macros and should complete within a few seconds using DREAPlace.
> >
> > As for Weakness 3, the reviewer acknowledges the challenge of fitting extensive information into a 9-page paper and appreciates the authors’ effort in balancing completeness and conciseness.

---

> > > ### Author Response · Authors · 2025-08-08
> > > **Response to Reviewer 9LHR.**
> > >
> > > Dear Reviewer 9LHR,
> > >
> > > Thank you for your kind support and for helping us improve the paper. We sincerely appreciate your valuable suggestions.
> > >
> > > We agree with your observation regarding the testcase limitations in our additional global placement experiments. In response, we have benchmarked the global placement on additional testcases from the ChiPBench. The results are presented below:
> > >
> > > | OpenROAD        |         |           |      |        |        | DREAMPlace      |         |            |      |        |         |
> > > |-----------------|---------|-----------|------|--------|--------|-----------------|---------|------------|------|--------|---------|
> > > | Algorithm       | WNS     | TNS       | NVP  | Power  | Area   | Algorithm       | WNS     | TNS        | NVP  | Power  | Area    |
> > > | dft68           | -0.292  | -57.656   | 275  | 0.237  | 88779  | dft68           | -0.578  | -47.246    | 261  | 0.083  | 97422.5 |
> > > | ethernet        | -0.199  | -10.524   | 176  | 0.118  | 96157  | ethernet        | -0.111  | -3.157     | 99   | 0.118  | 96052.6 |
> > > | VeriGPU         | -0.294  | -51.109   | 491  | 0.094  | 153084 | VeriGPU         | -0.311  | -23.777    | 300  | 0.093  | 152256  |
> > > | ariane81        | -0.298  | -397.887  | 2151 | 0.189  | 339853 | ariane81        | -2.134  | -2288.530  | 3334 | 0.148  | 287848  |
> > > | bp_fe38         | -1.998  | -808.689  | 1383 | 0.111  | 62706  | bp_fe38         | -2.008  | -311.881   | 1079 | 0.107  | 58805.9 |
> > > | bp_multi57      | -2.499  | -621.463  | 407  | 0.105  | 203747 | bp_multi57      | -1.860  | -3.381     | 2    | 0.103  | 197548  |
> > > | swerv_wrapper43 | -0.482  | -352.959  | 1222 | 0.256  | 225634 | swerv_wrapper43 | -0.693  | -640.200   | 1446 | 0.268  | 226462  |
> > >
> > >
> > > note that the OpenROAD baseline generally achieves better results because it employs RePlace with timing-driven mode, whereas the original DREAMPlace is not timing-driven.
> > >
> > > To address your query on runtime, we wish to clarify that the runtime in Table 4 does not represent the placement time alone. Instead, it is the total **evaluation time**. As shown in Figure 2, this reported time is the cumulative runtime of all subsequent physical design stages:
> > >
> > > **Evaluation Time = Global Placement + Legalization + Clock Tree Synthesis + Routing**
> > >
> > > This time represents the full cost of obtaining the final PPA metrics for a given placement solution using our benchmark. We will clarify this definition in the revised version of our appendix to avoid any confusion for future readers.
> > >
> > > Once again, we sincerely thank you for your insightful comments and kind support.
> > >
> > > Best regards,
> > >
> > > The Authors

---

> > > > ### Comment · Reviewer_9LHR · 2025-08-08
> > > >
> > > > Thanks for the update! The reviewer doesn't have further questions.

---

### Official Review · Reviewer_qNbE · 2025-06-29

**Rating:** 4
**Confidence:** 3

**Summary:**

ChiPBench introduces an open-source, end-to-end workflow to evaluate the effectiveness of different chip placement algorithms using PPA metrics, as opposed to intermediate surrogate metrics. Results indicates that there is significant gap between intermediate metrics and final PPA metrics.

**Additional Feedback:**

Questions
1.	To my understanding, the main contribution of this work is an open-source end-to-end workflow to evaluate the final PPA metrics. Prior works opted to use intermediate metrics for computational efficiency. What is the time of the workflow to compute final PPA metrics vs intermediate metrics? This would help potential users better understand the pros versus cons of the workflow.
2.	“For each stage, the dataset provides intermediate design data for every case, enabling tasks such as logic optimization, chip placement, and routing in the EDA domain.”  The results presented focus on chip placement. Would the authors provide a performance comparison of different algorithms for logic optimization and routing using the proposed dataset?
3.	The authors mention that commercial EDA tools are closed-source and expensive. How do the PPA metrics from the proposed open-source pipeline compare against the commercial counterparts? Why should I trust the results?

**Dataset Code Accessibility:**

Yes

**Ethical Considerations:**

No, there are no or only very minor ethics concerns

**Final Justification:**

I recommend that the authors include the additional results in the revised version to better contextualize the work and help users understand the runtime expectations. Overall, I believe the main contributions of this work are:
1. Reproducible evaluation framework for evaluating end-to-end chip placement algorithms
2. Highlight the gap between the surrogate metrics and end-to-end performance, paving the path for designing better surrogate metrics

More experiments can be included to better support the claim that the framework supports all stages of EDA. Existing experiments have only been conducted on placement and logic synthesis.

**Limitations Weaknesses:**

1.	Some claims are not well-supported and missing key details, please see Questions
2.	Lack of comparison against existing EDA workflows as sanity check, please see Questions

**Strengths Contributions:**

1.	The paper highlights the gap between intermediate surrogate metrics and final PPA using quantitative results, which calls for more careful selection of surrogate metrics for evaluating AI-based chip placement algorithms
2.	The writing is clear and easy to follow, providing sufficient context to understand the problem at hand
3.	The choice of metrics and baselines are reasonable
4.	Presents an open-source end-to-end workflow for computing PPA metrics, which is useful as an alternative to closed-source counterparts.

---

> ### Author Rebuttal · Authors · 2025-07-31
>
> Dear reviewer qNbE,
>
> Thank you for your insightful and valuable comments. We sincerely hope our rebuttal could adequately address your concerns. If so, we would deeply appreciate it if you could consider raising your score. If not, please let us know your further concerns, and we will continue actively responding to your comments.
>
>
> # Question 1. The time  to compute final PPA metrics vs intermediate metrics
>
> Thanks for your insightful comments.
> Our goal with ChiPBench is to highlight the potential misalignment of intermediate Surrogate metrics with end-to-end metrics and provide a framework to measure the final PPA.
>
> **Intermediate surrogate metrics** are typically calculated after specific stages in the flow. For example, metrics such as HPWL and MacroHPWL are reported after the placement stage.
>
> **Final PPA Metrics** (e.g., WNS, TNS, Power, Area) can only be accurately reported after running the entire physical design backend flow, which includes Clock Tree Synthesis  and Routing, in addition to placement.
>
> In our paper, we have provided the total evaluation runtime for each algorithm on each design in **Table 4 (Appendix C)**. This time represents the full end-to-end workflow required to obtain the **final PPA metrics**.For clarity, we provide the evaluation times for different intermediate and final PPA metrics across the selected designs in the table below.
>
> **Table1:Evaluation time (in seconds) for different  stage metrics**
>
> |               | Placement Metrics | Route Metrics | PPA Metrics |
> | ------------- | ----------------- | ------------- | ----------- |
> | ariane133     | 84                | 5958          | 6698        |
> | ariane136     | 62                | 7558          | 8258        |
> | bp            | 74                | 8888          | 11288       |
> | bp_be         | 100               | 22963         | 23766       |
> | bp_fe         | 83                | 5128          | 5423        |
> | swerv_wrapper | 74                | 30235         | 30723       |
>
>
>
> # Question 2. Performance comparison of different algorithms for other stages
>
> Thank you for your suggestion.
>
> You are correct in observing that our proposed dataset and workflow are designed to be extensible for evaluating other EDA stages, such as logic synthesis and routing.
>
> To demonstrate the capability of our framework beyond placement, we can show a comparative analysis at the logic synthesis stage. We used our dataset to compare a popular open-source synthesis tool (Yosys[1]) against a commercial EDA tool (Synopsys Design Compiler[2]). The results is below:
>
> The results are shown below:
>
> **Table 2: Final PPA Comparison Between Commercial and Open-Source Tools**
>
> | Case          | WNS (yosys) | WNS (DC)   | TNS (yosys) | TNS (DC)  | NVP (yosys) | NVP (DC) | Power (yosys) | Power (DC) | Area (yosys) | Area (DC) |
> |---------------|-------------|------------|-------------|-----------|-------------|----------|---------------|------------|--------------|-----------|
> | ariane133     | -0.129      | -0.052077  | -75.056     | -12.344   | 1321        | 402      | 0.342         | 0.3040677  | 389967       | 387311    |
> | ariane136     | -0.116      | -0.068622  | -50.597     | -21.342   | 1079        | 591      | 220.379       | 0.3254818  | 397302       | 396284    |
> | bp            | -4.822      | -4.121     | -22458.4    | -72.345   | 14758       | 621      | 0.488         | 0.496      | 525713       | 534217    |
> | bp_be         | -0.529      | -0.8497    | -39.002     | -41.63    | 110         | 144      | 0.416         | 0.1235193  | 122135       | 123221    |
> | bp_fe         | -0.116      | -0.056     | -1.007      | -0.711    | 32          | 28       | 0.167         | 0.14262    | 70658        | 72962     |
> | swerv_wrapper | -0.66       | -0.70343   | -613.774    | -697.845  | 1435        | 1500     | 0.265         | 0.2400538  | 226536       | 229277    |
>
>
> The comparison reveals that there is still a significant performance gap between existing open-source synthesis tools and mature commercial solutions. We believe that the proposed evaluation framework and dataset can play a vital role in accelerating the advancement of open-source synthesis tools.
>
>
>
>
>
> # Question 3. Lack of comparison of commercial EDA tools
>
>
> Thank you for your valuable question.
>
> Our motivation is to provide a **transparent, accessible, and reproducible** framework for the academic community. Commercial EDA tools are often "black boxes," making it difficult to reproduce results or understand the underlying heuristics. They also present significant access and cost barriers for many researchers.
>
> To validate the **credibility** of our open-source flow and address your trust question, we conducted a cross-verification experiment. We took the placement results generated by different AI algorithms and evaluated them using an industry-standard commercial toolchain (from Synopsys [3]). The goal was to check if the _relative ranking_ of the algorithms' quality is consistent between our open-source flow and the commercial flow.
>
>
>
> **Table 3: Comparison of Results Between Commercial EDA Tool Flow and Our Proposed Flow （after Normalized）**
>
> | Commercial EDA Tool |           |           |           |       | Our Flow     |           |           |           |       |
> | ------------------- | --------- | --------- | --------- | ----- | ------------ | --------- | --------- | --------- | ----- |
> | Algorithm           | WNS       | TNS       | NVP       | power | Algorithm    | WNS       | TNS       | NVP       | power |
> | AutoDMP             | 1.129     | 1.041     | 1.271     | 1.647 | AutoDMP      | 1.316     | 1.501     | 1.12      | 1.022 |
> | WireMask-EA         | 1.161     | 1.057     | 1.014     | 1.018 | WireMask-EA  | 1.28      | 1.58      | 1.039     | 1.033 |
> | ChiPFormer          | 1.132     | 1.057     | 1.014     | 1.018 | ChiPFormer   | 1.976     | 3.422     | 1.243     | 1.035 |
> | DREAMPlace          | 1.331     | 1.483     | 1.276     | 1.119 | DREAMPlace   | 1.56      | 2.259     | 1.037     | 1.018 |
> | MaskPlace           | 1.355     | 1.151     | 1.017     | 1.03  | MaskPlace    | 1.818     | 3.425     | 1.188     | 1.036 |
> | MaskRegulate        | **0.961** | **0.813** | **0.976** | 1.05  | MaskRegulate | **0.962** | **0.935** | **0.769** | 1.002 |
> | SA                  | 1.291     | 1.158     | 0.984     | 1.03  | SA           | 1.801     | 1.392     | 0.716     | 1.022 |
> | Hier-RTLMP          | 1         | 1         | 1         | **1** | Hier-RTLMP   | 1         | 1         | 1         | **1** |
>
>
> As shown in Table 3, the **relative performance ranking of the algorithms remains consistent**. Both our open-source flow This consistency demonstrates that **ChiPBench is a reliable and trustworthy framework for differentiating the quality of placement algorithms**, which is the core purpose of our benchmark.
>
>
>
> [1] Wolf, Clifford. "Yosys open synthesis suite." 3 Nov. 2016,
>
> [2] Compiler, Synopsys Design. "Synopsys design compiler." _Pages/default. aspx_ (2016).
>
> [3] Synopsys, Inc. "Electronic Design Automation (EDA)." Synopsys. Accessed: July 31, 2025.

---

> > ### Comment · Reviewer_qNbE · 2025-08-02
> >
> > Thank you for providing the additional experiments to address my concerns. I recommend that the authors include the additional results in the revised version to better contextualize the work and help users understand the runtime expectations. Overall, I believe the main contributions of this work are:
> > 1. Reproducible evaluation framework for evaluating end-to-end chip placement algorithms
> > 2. Highlight the gap between the surrogate metrics and end-to-end performance, paving the path for designing better surrogate metrics
> > More experiments can be included to better support the claim that the framework supports all stages of EDA. Existing experiments have only been conducted on placement and logic synthesis.
> >
> > In light of this, I have raised my score.

---

> > > ### Author Response · Authors · 2025-08-08
> > > **Thank you for your kind support.**
> > >
> > > Dear Reviewer qNbE,
> > >
> > > Thanks for your kind support and for helping us improve the paper. We sincerely appreciate your valuable suggestions.
> > >
> > > Best regards,
> > >
> > > The Authors

---

### Official Review · Reviewer_XWUw · 2025-07-01

**Rating:** 5
**Confidence:** 3

**Summary:**

The submission introduces ChiPBench, a comprehensive benchmark and evaluation framework targeting end-to-end chip placement using open-source tools and emphasizing physical design outcomes (e.g., Power, Performance, and Area) for evaluation. The authors build 20 chip designs from Verilog RTL and process them through an open-source physical design flow using OpenROAD, providing full reproducibility. They benchmark six state-of-the-art AI-based placement algorithms and traditional baselines, revealing critical discrepancies between intermediate and final metrics. The dataset, toolflow, and evaluation interface are all released publicly to support future research. As noted in the summary, the reviewer sees the technical contributions as: 1) A reproducible, open-source benchmark suite for chip placement, 2) A curated dataset of 20 designs with complete netlists, macros, timing constraints, and PPA evaluation reports, 3) A compelling empirical analysis showing that intermediate metrics poorly correlate with final PPA, motivating new proxy metric research.

**Additional Feedback:**

Including visual diagrams of design would make the dataset more accessible and easier to interpret for novices to the EDA space (but having ML experience).

**Dataset Code Accessibility:**

Yes

**Dataset Code Comments:**

The ChiPBench dataset is released on Hugging Face (https://huggingface.co/datasets/MIRA-Lab/ChiPBench-D), with supporting code and evaluation tools hosted at https://github.com/MIRALab-USTC/ChiPBench. The dataset includes Verilog sources, synthesis scripts, LEF/DEF files, and final PPA results, all formatted for plug-and-play use with the OpenROAD flow.

**Ethical Comments:**

The dataset is based entirely on synthetic hardware designs and open-source toolflows. No human data, privacy concerns, or licensing violations are present. The release follows community norms for EDA benchmarking and encourages transparency and reproducibility.

**Ethical Considerations:**

No, there are no or only very minor ethics concerns

**Final Justification:**

I am raising my score from 4 to 5 based on a clear and comprehensive rebuttal. The authors addressed all of my concerns thoughtfully and substantively. Specifically: (1) they contextualized the dataset scale relative to prior work and emphasized extensibility via automated tools; (2) they strengthened evaluation by including new commercial baselines using Synopsys tools; and (3) they committed to a Contribution Guide to support broader community use. While the benchmark does not yet support iterative co-optimization, the authors clearly articulated that their goal is evaluation infrastructure, not a training framework. That all being said, I still feel that expanding the initial dataset further will help build momentum and grow ChiPBench into a widely adopted and impactful community resource and so I urge the authors to build on this strong foundation soon.

**Limitations Weaknesses:**

Limitations/weaknesses can be broken down as follows:
- While the dataset is impressive, 20 designs remain limited relative to the full diversity of possible workloads
- Similarly, although six algorithms are benchmarked, additional baselines from commercial tools (even if with anonymized results) or ablation studies would add context to the strengths and weaknesses observed
- The framework evaluates post-placement PPA but does not yet support iterative co-optimization
- As the framework would benefit from community support, not having dedicated features and plans for how folks can provide additional data or link in their algorithms is a missed opportunity

**Strengths Contributions:**

As mentioned in the summary the strengths/contributions can be broken down as follows:
+ First fully open-source, end-to-end placement benchmark connecting source RTL to final PPA using reproducible flows (via OpenROAD)
+ Offers 20 diverse chip designs from CPUs, NPUs, and memory controllers, synthesized with macro diversity and real-world constraints
+ Benchmarks six state-of-the-art placement algorithms and reports rich intermediate and final metrics
+ Provides a critical empirical insight that common surrogate metrics like congestion have weak correlation with timing-driven PPA, underscoring the need to rethink evaluation metrics in the computer architecture community

---

> ### Author Rebuttal · Authors · 2025-07-31
>
> Dear reviewer XWUw,
>
> Thank you for your insightful and valuable comments. We sincerely hope our rebuttal could adequately address your concerns. If so, we would deeply appreciate it if you could consider raising your score. If not, please let us know your further concerns, and we will continue actively responding to your comments.
>
> # Weakness 1
>
> > While the dataset is impressive, 20 designs remain limited relative to the full diversity of possible workloads.
>
> Thank you for this valid point. We would like to clarify that our contribution's primary focus is on **quality, end-to-end realism, and extensibility**, which we believe are more urgent needs than sheer quantity.
>
> First, our set of 20 designs already marks a **significant expansion** over the academic benchmarks commonly used in placement algorithm papers, such as the 8-circuit ICCAD 2015 suite [1]. Crucially, unlike previous work, we provide the **complete RTL-to-GDSII context** (Verilog, SDC, LEF), enabling end-to-end evaluation.
>
> Second, and most importantly, ChiPBench is designed as an **extensible, living platform**, not a static dataset. Our open-source and automated generation pipeline (Sec. 4.3) allows any researcher to create new, diverse benchmarks with ease. This design choice directly empowers the community to expand the dataset, ensuring its long-term relevance and addressing the very concern of limited diversity.
>
>
>
>
> # Weakness 2
>
> > Although six algorithms are benchmarked, additional baselines from commercial tools (even if with anonymized results) or ablation studies would add context to the strengths and weaknesses observed.
>
> Thank you for this excellent suggestion. We agree that comparing our results with commercial EDA tools would provide valuable context.We have conducted this analysis and plan to include the following table and discussion in the final version of our paper. We performed placement on 6 designs from our dataset using commercial EDA tools (Synopsys [2]), and then evaluated the results using our own flow.
> (Due to time constraints, we did not run this test on all designs.) The results are as follows:
>
> **Table1:Evaluation Results Using Synopsys Commercial Tool**
>
> |                   | **MacroHPWL($\mu m$)** | **Regularity($\mu m$)** | **HPWL($\mu m$)** | **Congestion** | **Wirelength($\mu m$)** | **WNS(ns)** | **TNS(ns)** | **NVP** | **Power(nW)** | **Area($\mu m^{2}$)** |
> | ----------------- | ---------------------- | ----------------------- | ----------------- | -------------- | ----------------------- | ----------- | ----------- | ------- | ------------- | --------------------- |
> | **ariane133**     | 180241                 | 118308                  | 6077457.9         | 0.244          | 7264950                 | -0.0520     | -12.34      | 402     | 0.304         | 387311                |
> | **ariane136**     | 186767                 | 117957                  | 6130882           | 0.248          | 7355342                 | -0.0686     | -21.34      | 591     | 0.325         | 396284                |
> | **bp**            | 23048                  | 17401                   | 8401960           | 0.515          | 944277                  | -4.12       | -72.34      | 621     | 0.496         | 534217                |
> | **bp_be**         | 18821                  | 3681                    | 3346075           | 0.607          | 4150867                 | -0.849      | -41.63      | 144     | 0.123         | 123221                |
> | **bp_fe**         | 36544                  | 4110                    | 2356003           | 0.583          | 3161588                 | -0.0562     | -0.7112     | 28      | 0.142         | 72962                 |
> | **swerv_wrapper** | 96692                  | 12794                   | 4143879           | 0.355          | 4970401                 | -0.703      | -697.8      | 1500    | 0.240         | 229277                |
>
> **Table2:Normalized Evaluation Results Compared to the Baseline**
>
> |                   | **MacroHPWL** | **Regularity** | **HPWL** | **Congestion** | **Wirelength** | **WNS** | **TNS** | **NVP** | **Power** | **Area** |
> | ----------------- | ------------- | -------------- | -------- | -------------- | -------------- | ------- | ------- | ------- | --------- | -------- |
> | **ariane133**     | 0.317         | 1.798          | 0.926    | 0.919          | 0.944          | 0.404   | 0.164   | 0.304   | 0.889     | 0.993    |
> | **ariane136**     | 0.355         | 1.586          | 0.959    | 0.947          | 0.970          | 0.592   | 0.422   | 0.548   | 0.859     | 0.997    |
> | **bp**            | 1.019         | 1.228          | 1.049    | 1.232          | 0.101          | 0.855   | 0.003   | 0.042   | 1.016     | 1.016    |
> | **bp_be**         | 1.386         | 0.850          | 1.064    | 1.028          | 1.030          | 1.606   | 1.067   | 1.309   | 0.846     | 1.009    |
> | **bp_fe**         | 0.751         | 1.007          | 1.122    | 1.214          | 1.203          | 0.483   | 0.706   | 0.875   | 0.854     | 1.033    |
> | **swerv_wrapper** | 0.818         | 0.765          | 1.089    | 1.091          | 1.092          | 1.066   | 1.137   | 1.045   | 0.906     | 1.012    |
> | **Avg.**          | 0.774         | 1.205          | 1.035    | 1.072          | 0.890          | 0.834   | 0.583   | 0.687   | 0.895     | 1.010    |
>
> These results show that, compared to other AI-based algorithms reported in the paper, commercial tools still hold a certain advantage. This highlights the need for continued development within the open-source EDA community, particularly in optimizing for PPA.
>
> # Weakness 3
>
> > The framework evaluates post-placement PPA but does not yet support iterative co-optimization.
>
>
> Thank you for this important observation. We would like to emphasize that our work does not propose a new placement algorithm or a co-optimization loop. The primary contribution of our work is to propose a dataset and evaluation flow to highlight the critical disconnect between current placement algorithms and final design metrics, with the hope of encouraging existing placement research to focus more on PPA optimization. Due to the lack of open-source evaluation tools in this area, ChiPBench is introduced to help fill this gap.
>
>
>
> # Weakness 4
>
> > - As the framework would benefit from community support, not having dedicated features and plans for how folks can provide additional data or link in their algorithms is a missed opportunity
>
> Thank you for your concern. We completely agree that community contribution is vital for the long-term success of ChiPBench.To address this concretely, we will add a **Contribution Guide** to our public GitHub repository. This document will formalize the process for community collaboration.  We will provide detailed instructions on how to use our project to generate **additional data** using our open-source generation scripts, and how to **report and submit** these results to us. We believe this will foster a collaborative ecosystem around EDA research.
>
>
> [1] Kim, Myung-Chul, et al. "ICCAD-2015 CAD contest in incremental timing-driven placement and benchmark suite." _2015 IEEE/ACM International Conference on Computer-Aided Design (ICCAD)_. IEEE, 2015.
> [2] Synopsys, Inc. "Electronic Design Automation (EDA)." Synopsys. Accessed: July 31, 2025.

---

> > ### Comment · Reviewer_XWUw · 2025-08-05
> > **Response to Rebuttal**
> >
> > Thank you for the detailed and thoughtful rebuttal. The added commercial baselines, clarification of benchmark scope, and commitment to supporting community contributions meaningfully strengthen the paper. I appreciate your responsiveness and believe ChiPBench now offers a valuable and timely infrastructure contribution to the EDA and ML-for-systems communities. That all being said, I still feel that further expanding the initial dataset will enable it to begin the snowball effect of growing to a very exciting project. So I urge the authors to build on this work soon.

---

> > > ### Author Response · Authors · 2025-08-08
> > > **Response to Reviewer XWUw.**
> > >
> > > Dear Reviewer XWUw,
> > >
> > > Thank you for your kind support and for helping us improve the paper. We sincerely appreciate your valuable suggestions.
> > >
> > > You are exactly right about expanding the dataset to help the project grow—like a "snowball effect." This is a key goal for us. To achieve this, we built our open-source tool that automatically generates new benchmarks (as described in Sec. 4.3). Our aim is to make it simple for us and for others in the community to add a wide variety of new cases.
> > >
> > > We will certainly follow your advice and continue to build on this by adding more diverse benchmarks in our upcoming work.
> > >
> > > Once again, we sincerely thank you for your insightful comments and kind support.
> > >
> > > Best regards,
> > >
> > > The Authors

---

### Official Review · Reviewer_Xvde · 2025-07-01

**Rating:** 5
**Confidence:** 3

**Summary:**

In this work, the authors provide benchmarks for chip placement algorithms. More specifically, the proposed framework enables the end-to-end evaluation of AI-based algorithms for chip placement. The benchmark incorporates various phases of electronic design automation, ranging from placement to routing. The authors integrate different open-source tools, such as OpenROAD and NanGate45, to construct a pipeline, which is provided as an easy-to-use benchmark on the Hugging Face platform. According to authors the proposed benchmark is designed to evaluate the effectiveness of AI-based algorithms in final design Performance-Power-Area metrics allowing to overcome the existing misalignment with intermediate surrogate metrics.

**Dataset Code Accessibility:**

Partly

**Ethical Considerations:**

No, there are no or only very minor ethics concerns

**Final Justification:**

The work proposes an end-to-end evaluation framework which is crucial for designing intermediate metrics that are better aligned with final PPA and better learning to balance cost and accuracy in AI-based placement algorithms. Even though the practicality of such contribution is undoubtable, I have pointed pointing out the framework can not be effectively used for end-to-end training, utilizing final PPA metric in an iterative optimization process.

The authors extensively discuss the aforementioned limitation, while they address my other concerns, improving the usability of the framework by enhancing the documentation and adding Contribution Guide.

**Limitations Weaknesses:**

Although an end-to-end evaluation metric for AI-based placement algorithms sounds promising, it seems impractical for training without utilizing the surrogate metrics, especially for gradient based optimization.

Even though the authors report a correlation analysis between intermediate metric and finally PPA metrics, they do not utilize it to introduce a metric that can be effectively used by placement algorithms. This limits the contribution of this work, since, from my understanding, a single end-to-end evaluation introduces a significant performance and time bottleneck.

The authors should better define what Evaluation Time means in Tables C.4 and C.5.

The code provided with the benchmark is far from being easily usable. It is merely documented and includes huge batches of commented code.

**Strengths Contributions:**

The paper is very well written, it provides the necessary background information and presents in detail the pipeline used to generate the proposed benchmark. Additionally, the proposed benchmark utilizes open-source tools and the authors provide the benchmark through Hugging Face, which allows one to easily use it.

The benchmarks provide 20 different designs for different levels of difficulty while the open source nature of the utilized tools ensures the accessibility and extensibility.

The end-to-end approach seems promising when it comes to the evaluation of AI-based algorithms. Furthermore, the authors present comprehensive experimental evaluation for the baselines, reporting metrics both related to the effectiveness of the proposed method (such as metrics for placement, route timing, power and area) and efficiency of each method (Evaluated Time and Memory Usage).

Finally, the paper presents a correlation analysis between intermediate metric and finally PPA metrics, revealing the limitations of the former ones, verifying the motivation of the authors.

---

> ### Author Rebuttal · Authors · 2025-07-31
>
> Dear reviewer Xvde,
>
> Thank you for your insightful and valuable comments. We sincerely hope our rebuttal could adequately address your concerns. If so, we would deeply appreciate it if you could consider raising your score. If not, please let us know your further concerns, and we will continue actively responding to your comments.
>
> # Weakness : Practicality of End-to-End Evaluation
>
>
> Thank you for this very insightful comment.
>
> We acknowledge the concern that while we analyze the correlation between intermediate metrics and final PPA, we do not introduce a new, fully validated surrogate metric. However, the primary contribution of our work is to propose a dataset and evaluation flow to **highlight the critical disconnect between intermediate surrogate metrics  and final design metrics**, rather than to propose a new, immediately usable surrogate metric ourselves. Our goal is first to **diagnose** the problem before attempting to solve it.
>
> Our evaluation of different placement methods already provides valuable insights into which metrics show promise as effective surrogates. For example, our correlation analysis in Figure 3 reveals that the **Regularity** metric, used by MaskRegulate [4], demonstrates a notably stronger correlation with final timing performance compared to other common intermediate metrics. This is an important clue for future research.
>
> Indeed, our work is intended to pave the way for these next steps. As we discuss in **Section 6.4**, we believe our benchmark will enable the community to pursue critical future research directions, including:
>
> - Designing intermediate metrics that better align with final PPA by organically integrating factors such as regularity and dataflow;
> - Developing feature-based surrogate models to approximate final PPA more efficiently;
> - Leveraging multi-fidelity optimization and learning to balance cost and accuracy in AI-based placement algorithms.
>
> Crucially, all of these promising research avenues are fundamentally dependent on having a reliable **end-to-end evaluation framework** to validate their effectiveness, which our work provides.
>
>
>
> # Weakness : Definition of Evaluation Time
>
> Thank you for pointing out the need for a clearer definition.
>
> The **"Evaluation Time"** reported in Table 4 (Appendix C) refers to the runtime of our  **evaluation flow**. We start the clock after a given macro placement algorithm has produced its output. The reported time is the sum of the runtimes for all subsequent stages in the physical design flow, as illustrated in our **Figure 2**. Specifically, it includes:
>
> **Evaluation Time = Global Placement + Legalization + Clock Tree Synthesis + Routing**
>
> This time represents the full cost of obtaining the final PPA metrics for a given placement solution using our benchmark. We will clarify this definition in the revised version of our appendix to avoid any confusion for future readers.
>
> # Weakness : Code Documentation and Usability
>
> Thank you for your valuable feedback on our code repository. We take reproducibility and ease-of-use very seriously, and we appreciate you bringing this to our attention. Your point is well-taken.
>
> We have updated the repository as follows:
>
> 1. **Enhanced Documentation:** We modified document with step-by-step instructions for setting up the environment, running the evaluation scripts.
> 2. **Code Refactoring:** We removed large blocks of commented-out code to improve clarity and maintainability.
>
> We hope these updates will make the tool significantly easier to use.

---

> > ### Comment · Reviewer_Xvde · 2025-08-05
> >
> > I would like to thank the authors for answering my concerns.
> >
> > The contribution of this work is indeed focused on diagnosing problems that are associated with the misalignment between intermediate surrogate metrics and final design metrics. The work proposes an end-to-end evaluation framework which is crucial for designing intermediate metrics that are better aligned with final PPA and better learning to balance cost and accuracy in AI-based placement algorithms. Even though the practicality of such contribution is undoubtable, in my first comment of the original review I am pointing out the framework can not be effectively used for end-to-end training, utilizing final PPA metric in an iterative optimization process. In my opinion, such a discussion it might be useful for future works.
> >
> > Nevertheless, the authors address my other concerns, improving the usability of the framework by enhancing the documentation and adding Contribution Guide as it is suggested by Reviewer XWUw.
> >
> > I am willing to keep my score for now.

---

> > > ### Author Response · Authors · 2025-08-08
> > > **Thank you for your kind support.**
> > >
> > > Dear Reviewer Xvde,
> > >
> > > Thank you for your kind support and for helping us improve the paper. We sincerely appreciate your valuable suggestions.
> > >
> > > We agree with your main concern about the impracticality of using our framework for direct, iterative end-to-end training. That said, we want to reaffirm that the core purpose of our work is to **provide an evaluation method that reveals the AI community’s overemphasis on surrogate metrics like HPWL**. Consequently, innovative algorithms that may perform poorly on such simple proxies but achieve better final PPA are unfairly penalized and overlooked. Our benchmark is meant to be a **fair judge**, offering a platform where truly effective methods can demonstrate their real advantages.
> > >
> > > More importantly, our framework is built to **help drive the research needed to solve the training challenges you mentioned.** It provides a solid foundation for future work in several important areas:
> > >
> > > 1. **Improving Surrogate Metrics:** Our flow provides the **ground-truth PPA labels** needed to train and carefully evaluate new surrogate metrics. With this, researchers can explore better proxies (e.g., ones that consider regularity, as we mentioned) and use our benchmark to test how well these proxies correlate with final PPA results.
> > > 2. **Supporting Data-Driven Methods:** Our evaluation process enables the creation of high-quality `(placement, final PPA)` datasets. This opens the door for powerful **offline learning methods**, such as offline reinforcement learning or imitation learning, which avoid the cost of online training.
> > >
> > > We will include a discussion on this in the final revision to help guide future research on improving training methods.
> > >
> > > Once again, we sincerely thank you for your insightful comments and kind support.
> > >
> > > Best regards,
> > >
> > > The Authors

---

### Note · Authors · 2025-08-13

Dear Area Chair and Reviewers,

We sincerely thank the reviewers for their valuable comments and constructive suggestions on our work.

Our paper has received encouraging comments from the reviewers, highlighting various positive aspects, including ChiPBench’s value as a reproducible, open-source, end-to-end evaluator for EDA flow ,supported by a diverse benchmark, lowers barriers to benchmarking, and provides credible, standardized measurements.

In response, we have addressed each point in detail and conducted extensive additional experiments to further improve the quality of our research.

Specifically, during the rebuttal we added the following experiments and analyses to carefully address the reviewers’ questions:
- Added commercial baselines (Synopsys) on representative designs and reported normalized results.
- Cross-checked a subset of results using a commercial toolchain and found that the relative ranking of algorithms remains consistent between our open-source flow and the commercial flow.
- Used our evaluation flow and benchmark to evaluate logic synthesis algorithms (Yosys and Synopsys DC) and global placement algorithms (RePlace and DREAMPlace), and reported final PPA to address the reviewers’ concerns on extensibility.
- Reported comparative runtimes for computing final PPA metrics versus intermediate metrics.
- Clarified the definition of Evaluation Time in Table 4 (Appendix C).

Moreover, we incorporated the reviewers’ valuable suggestions, such as:
- Enhancing documentation with step-by-step instructions, adding a Contribution Guide, and refactoring the repository by removing large commented blocks to make extension and adoption easier.
- Improving the writing of the paper.

Additionally, we reiterate that our intent is to highlight the disconnect between intermediate metrics and final PPA, and to provide a solid foundation for future work that maintains alignment with final PPA.

We truly appreciate the reviewers' time and effort in providing insightful feedback, which has significantly contributed to strengthening our work.

Best regards,
Authors

---

### Decision · Program_Chairs · 2025-09-18

**Decision:**

Accept (poster)

**Comment:**

## Summary:
This paper introduces ChiPBench, a comprehensive, open-source benchmark designed to evaluate AI-based chip placement algorithms. The central thesis is that the academic community's focus on intermediate surrogate metrics (e.g., half-perimeter wirelength) is misaligned with the final, industry-relevant metrics of performance, power, and area (PPA). To address this, the authors provide an end-to-end evaluation pipeline using open-source Electronic Design Automation (EDA) tools, accompanied by a diverse dataset of 20 circuit designs. Their experiments on six state-of-the-art AI placement algorithms reveal a significant disconnect between the surrogate metrics these algorithms optimize and the final PPA results, underscoring the benchmark's value.

## Pros:
- Significance and Novelty: The paper addresses a critical and well-recognized gap between academic research in AI for EDA and industrial practice. It provides the first fully open-source, reproducible, end-to-end benchmarking framework that connects high-level design files (RTL) to final PPA metrics.
- High-Quality Resource: The submission provides a valuable and accessible resource to the community, including a diverse dataset of 20 designs, the full evaluation toolchain, and benchmark results, all hosted publicly on Hugging Face and GitHub. This significantly lowers the barrier for researchers to perform realistic and rigorous evaluations.
- Thorough Evaluation and Analysis: The authors benchmark six different AI-based algorithms and provide a compelling correlation analysis that empirically validates the paper's core motivation—the inadequacy of common surrogate metrics.
- Responsiveness and Improvement during Rebuttal: The authors provided an exemplary rebuttal, conducting substantial new experiments to address every major concern raised by the reviewers. This included adding comparisons to a commercial EDA tool (Synopsys), demonstrating the framework's extensibility to other EDA tasks like logic synthesis and global placement, and providing detailed runtime analyses.

## Cons:
- Evaluation, Not Training: The primary weakness, acknowledged by the authors, is that the end-to-end PPA evaluation is too computationally expensive to be used directly as a reward signal for iterative training of AI agents (e.g., via reinforcement learning). The work is thus an evaluation framework, not a training solution.
- Initial Scope and Usability Issues: The initial submission was primarily focused on macro placement, and the clarity and documentation of the code needed improvement. However, these points were thoroughly addressed during the author-reviewer discussion phase.

## Recommendation:
I recommend this paper for acceptance. The authors have successfully identified a critical weakness in the current methodology for evaluating AI-based chip placement and have delivered a high-quality, practical solution. The work is a perfect fit for the Datasets and Benchmarks track.
My decision is heavily influenced by the authors' outstanding engagement during the rebuttal period. They went above and beyond to address the reviewers' concerns by:
- Adding Commercial Baselines: They benchmarked against a commercial Synopsys tool, contextualizing the performance of the open-source and AI-based methods.
- Validating the Open-Source Flow: They showed that the relative ranking of algorithms is consistent between their open-source flow and the commercial toolchain, building trust in their methodology.
- Demonstrating Extensibility: They ran new experiments to show the framework's utility for other EDA stages, such as logic synthesis and global placement, addressing the concern about the limited scope.
- Improving Usability: They significantly improved the documentation and code quality based on reviewer feedback.

The discussion period was highly productive. Initially, reviewers raised several key points: the lack of commercial baselines (XWUw, qNbE); the limited scope, focusing mainly on macro placement (9LHR); the impracticality for direct training (Xvde); poor code documentation (Xvde); and the need for runtime analysis (qNbE)
The authors addressed each point comprehensively. They added experiments with Synopsys tools, tested a global placer (DREAMPlace), evaluated logic synthesis tools, provided runtime breakdowns, and committed to improving documentation and community contribution guidelines.


This thorough response turned the paper into a good one, satisfying nearly all reviewer concerns and leading two reviewers (XWUw and qNbE) to raise their scores. The final paper represents a solid, well-vetted, and valuable contribution that will serve as a foundational resource for future research in this area.